# HAPDA: A HUMAN-MACHINE PREDICTIVE DISCREPANCY ADAPTER FOR AI-GENERATED TEXT DETECTION

## ABSTRACT

Recent advances in large language models (LLMs) have enabled them to generate text with increasingly human-like linguistic styles, posing significant challenges for AI-generated text detection (AGTD). Mainstream zero-shot AGTD methods primarily compute token-level AI-likeness scores from a machine-centric perspective represented by proxy models, and treat all tokens equally in the overall detection score calculation. However, these methods overlook predictive discrepancies between humans and LLMs in interpreting the same text. Our key intuition is that tokens exhibiting greater divergence in human and machine predictions offer stronger cues for authorship attribution. To address this limitation, we propose **HAPDA**, a human-machine predictive discrepancy adapter for the AGTD task. HAPDA consists of (i) a joint fine-tuning strategy for training paired human and machine preference models, and (ii) a discrepancy-aware reweighting mechanism to calibrate token-level detection scores in downstream detectors. Extensive experiments across multiple datasets demonstrate that HAPDA consistently and significantly improves the performance of five representative baselines under diverse evaluation settings.

## 1 INTRODUCTION

In recent years, LLMs have made remarkable strides, substantially enhancing the capabilities of natural language processing technologies. The text generated by these models now rivals human writing in fluency and coherence (Li, 2025). However, as LLMs see wider adoption in real-world applications, their associated risks have become increasingly evident, including challenges in verifying content authenticity (Zhang et al., 2025a), the spread of misinformation (Yin et al., 2024), and potential misuse (Abdali et al., 2024). Consequently, developing efficient and accurate methods for detecting AI-generated text has become critically important.

A class of zero-shot AGTD methods computes token-level "AI-likeness" scores (e.g., entropy (Gehrmann et al., 2019), probability (Solaiman et al., 2019)) and averages them to obtain a document-level score, classifying the text as AI-generated based on a predefined threshold. In our work, we refer to this family of methods as **MeanZero**. They offer the advantages of being fast, convenient and requiring no labeled training data, but their accuracy degrades when dealing with high-quality text generated by LLMs.

Existing research has increasingly recognized that not all tokens in a text contribute equally to identifying the author's identity. The discriminative power of low-probability tokens is highlighted by POGER (Shi et al., 2024); necessary tokens are extracted by PECOLA (Liu et al., 2024) using YAKE (Campos et al., 2020); textual coherence is modeled by CoCo (Liu et al., 2023) using a graph structure; and dynamic token weights based on semantics, context, and positional information are assigned by PAWN (Miralles-González et al., 2025). However, these methods are often based on supervised learning, which incurs high training costs and is prone to overfitting to specific training data. In addition, although MeanZero methods are enhanced using log-rank information by DetectLRR and DetectNPR (Su et al., 2023), they still operate solely from the LLMs' perspective and overlook the human generation mechanism.

Within the same passage, there exist significant differences in the predicted probability distributions of specific tokens between humans and machines (Ippolito et al., 2020); such differences can be regarded as the informational association between the token and the latent variable of "author identity." Besides, if a token's generation probability distribution differs greatly between human and machine models, it indicates higher **mutual information** for inferring author identity (Yoo et al., 2024; West et al., 2025; Zhang et al., 2025b). These tokens contribute more effectively to reducing uncertainty in the probabilistic space and are thus the most discriminative components. As suggested in Figure 1, our intuition is that *tokens exhibiting greater divergence in human and machine predictions offer stronger cues for authorship attribution*. We provide a mathematical analysis of this intuition from the perspective of mutual information in Appendix D.

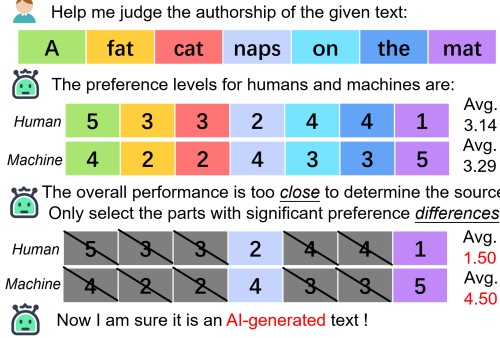

Figure 1: Illustration of human-machine predictive discrepancy in AGTD tasks. For the same piece of text, humans and machines may make similar predictions for some tokens. In such cases, tokens with significant predictive discrepancies often carry more information for authorship attribution.

In this work, we propose **HAPDA**, a framework designed to enhance the performance of MeanZero detectors. Motivated by Direct Preference Optimization (DPO) (Rafailov et al., 2023), we introduce a novel joint fine-tuning strategy called **HAPDA-Finetune**, which trains two auxiliary models based on an open-source language model. These models are optimized to exhibit stronger preferences for human-written and AI-generated texts, allowing them to capture predictive discrepancies between human and machine perspectives. Based on these auxiliary models, we propose a token-level reweighting mechanism, termed **HAPDA-Calibration**, which leverages token-wise disagreement and uncertainty to assign higher weights to more discriminative tokens during the computation of the overall detection score. Our main contributions are summarized as follows:

- We are the first to reconsider the AGTD task from a joint human and machine predictive perspective.
- We design a novel joint fine-tuning strategy that obtains a pair of LLMs with stronger preferences for human-written and AI-generated texts, respectively.
- We model the predictive discrepancies between human and machine to provide token-level reweighting for detection scores in MeanZero detectors.
- Extensive experiments across multiple datasets demonstrate that HAPDA consistently and significantly improves the performance of five representative baselines under diverse evaluation settings.

## 2 RELATED WORK

**AI-generated Text Detection.**  A common approach in AGTD exploits LLMs' tendency to generate tokens with higher conditional probabilities, reflecting greater "confidence." Metrics such as perplexity (Hans et al., 2024) and log probability (Xu et al., 2024b) have been used to identify AI-generated text. Mitchell et al. (2023) show that perturbations reduce these probabilities, while other studies (Shi et al., 2024; Wang et al., 2023) treat token-level probability sequences as features for supervised detection. LLM text also often exhibits lower entropy (Gehrmann et al., 2019), which has been used in watermarking to select insertion points (Wu et al., 2025; Liu & Bu, 2024) and adjust watermark strength (Lu et al., 2024). Our work improves zero-shot AGTD by amplifying tokens with large prediction discrepancies between human and machine models.

**Fine-tuning Strategies in AGTD Research.**  Fine-tuning has been widely used in AGTD. For example, Li et al. (2024) fine-tunes a rewriting model to amplify perturbations, Wang et al. (2025) uses reinforcement learning to humanize outputs of small LMs, Zeng et al. (2024) fine-tunes proxy

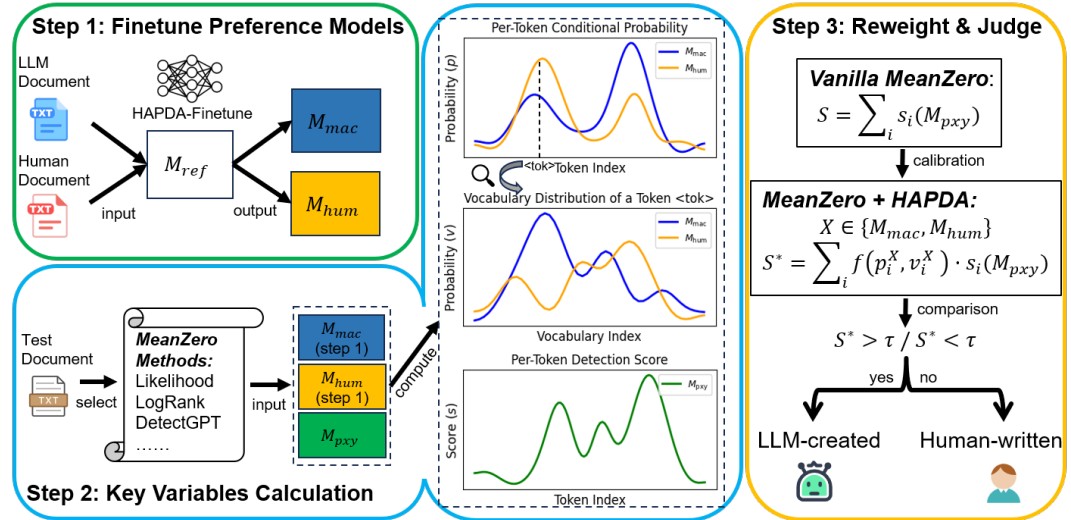

Figure 2: The workflow of HAPDA. Step 1: Fine-tune the machine-preference model $\mathcal{M}_{\mathrm{mac}}$ and the human-preference model $\mathcal{M}_{\mathrm{hum}}$ on a training corpus that contains both AI-generated and human-written texts. Step 2: Scan the input text to compute key variables. $\mathcal{M}_{\mathrm{mac}}$ and $\mathcal{M}_{\mathrm{hum}}$ are used to compute the per-token conditional probability $p$ and vocabulary distribution $v$, while the proxy model $\mathcal{M}_{\mathrm{pxy}}$ is used to compute the raw detection score of the selected MeanZero method. Step 3: Adjust the raw detection score using a reweighting coefficient derived from $p$ and $v$, and compare the adjusted score against a predefined threshold $\tau$ to determine the authorship of the given text.

models to better match source distributions, and Xu & Sheng (2024) adapts DetectGPT (Mitchell et al., 2023) to code generation. Our work focuses on fine-tuning auxiliary models to capture human–machine preferences.

# 3 PROPOSED METHOD

## 3.1 TASK DESCRIPTION

We consider the AGTD task from a document composed of tokens $\{x_1, x_2, \ldots, x_n\}$. A class of methods, referred to as MeanZero in this work, computes a detection score for each token $s(x_i)$, and aggregates them into a document-level score via simple averaging $S = \sum_{i=1}^{n} s(x_i)/n$, which is compared against a threshold to determine the text's authorship.

We generalize this framework by introducing a token-wise weighting mechanism. Rather than treating all tokens equally, we assign each token $x_i$ an importance weight $w_i$, yielding a weighted detection score $S^* = \sum_{i=1}^{n} w_i \cdot s(x_i)$. Therefore, we propose HAPDA, consisting of two components: HAPDA-Finetune (step 1 in Figure 2) and HAPDA-Calibration (steps 2 and 3 Figure 2).

## 3.2 HAPDA-FINETUNE

The goal of HAPDA-Finetune is to fine-tune a pair of models—a *Human-preference Model* ($\mathcal{M}_{\mathrm{hum}}$) and a *Machine-preference Model* ($\mathcal{M}_{\mathrm{mac}}$)—from a white-box *reference model* ($\mathcal{M}_{\mathrm{ref}}$), in order to achieve: (**Target 1**) *Alignment*: $\mathcal{M}_{\mathrm{hum}}$ should assign higher scores to human-written texts, while $\mathcal{M}_{\mathrm{mac}}$ should favor AI-generated texts; (**Target 2**) *Distinctiveness*: Under the same input, the scoring behaviors of $\mathcal{M}_{\mathrm{hum}}$ and $\mathcal{M}_{\mathrm{mac}}$ should diverge on both human and machine responses.

**Alignment: Guides the models to maintain consistent preferences for texts from specific sources, thereby enhancing the base discriminative power of detection.** To achieve **Target 1**, we adopt the DPO strategy proposed by Wang et al. (2025). DPO has been widely applied in LLMs with human preferences in response generation tasks. In our work, we adapt DPO to simultaneously fine-tune two models with opposing preferences: one that favors machine text and another that fa-

vors human text. Given a prompt $P$, a human-written response $H$, and a AI-generated response $M$, we define the alignment loss for models $\mathcal{M}_{\text{mac}}$ and $\mathcal{M}_{\text{hum}}$ as:

$$\mathcal{L}_{\text{ali}} = -\log \sigma(\Delta_{\text{mac}}) - \log \sigma(\Delta_{\text{hum}}),$$

where $\sigma(\cdot)$ is the sigmoid function. $\Delta_{\text{mac}}$ and $\Delta_{\text{hum}}$ denote the preference margins of $\mathcal{M}_{\text{mac}}$ and $\mathcal{M}_{\text{hum}}$, which are expressed as:

$$\Delta_{\text{mac}} = \log \frac{\pi_{\text{mac}}(M \mid P)}{\pi_{\text{ref}}(M \mid P)} - \log \frac{\pi_{\text{mac}}(H \mid P)}{\pi_{\text{ref}}(H \mid P)}, \quad \Delta_{\text{hum}} = \log \frac{\pi_{\text{hum}}(H \mid P)}{\pi_{\text{ref}}(H \mid P)} - \log \frac{\pi_{\text{hum}}(M \mid P)}{\pi_{\text{ref}}(M \mid P)},$$

where $\pi(\cdot \mid \cdot)$ denotes the conditional probability. For $\mathcal{M}_{\text{mac}}$, $M$ is treated as the preferred sample and $H$ as the less preferred one; the opposite holds for $\mathcal{M}_{\text{hum}}$. A lower $\mathcal{L}_{\text{ali}}$ indicates that the preference models are effectively aligned with their intended preference directions. As proven by Rafailov et al. (2023), DPO encourages preference alignment while controlling the deviation from the reference model.

**Distinctiveness: Encourages the models to produce different scoring behaviors on the same input, enhancing the separability between human and machine texts.** To achieve **Target 2** while maintaining alignment, we introduce a Jensen-Shannon (JS) divergence-based loss (Lin, 2002) to enhance the token-level distributional distinctiveness between $\mathcal{M}_{\text{hum}}$ and $\mathcal{M}_{\text{mac}}$. Specifically, for a given prompt $P$ and a response $Y \in \{M, H\}$, we define:

$$p_{\text{mac}}^{Y} = \text{softmax}\big(\log \pi_{\text{mac}}(Y \mid P)\big), \quad p_{\text{hum}}^{Y} = \text{softmax}\big(\log \pi_{\text{hum}}(Y \mid P)\big),$$

where $p_{\text{mac}}^{Y}$ and $p_{\text{hum}}^{Y}$ represent the token-level probability distributions from $\mathcal{M}_{\text{mac}}$ and $\mathcal{M}_{\text{hum}}$, respectively. Then we define the mixture distribution and KL divergence (Kullback & Leibler, 1951):

$$m^{Y} = \frac{1}{2}(p_{\text{mac}}^{Y} + p_{\text{hum}}^{Y}), \quad \text{KL}(p \parallel m) = \sum_{i} p_i \log \frac{p_i}{m_i}.$$

The JS divergence is expressed as :

$$\text{JS}(p_{\text{mac}}^{Y} \parallel p_{\text{hum}}^{Y}) = \frac{1}{2}\text{KL}(p_{\text{mac}}^{Y} \parallel m^{Y}) + \frac{1}{2}\text{KL}(p_{\text{hum}}^{Y} \parallel m^{Y}).$$

Finally, the distinctiveness loss is then defined as:

$$\mathcal{L}_{\text{dis}} = 2\log 2 - \text{JS}(p_{\text{mac}}^{M} \parallel p_{\text{hum}}^{M}) - \text{JS}(p_{\text{mac}}^{H} \parallel p_{\text{hum}}^{H}),$$

which encourages the two models to produce more distinguishable token distributions under the same input. Note that $\mathcal{L}_{\text{dis}} \in [0, 2\log 2]$, where a lower value indicates the preference models exhibit more distinct behaviors on the same input. Compared to adopting KL divergence as an alternative (Huang et al., 2024), JS divergence has the advantages of being symmetric, bounded, and more stable during optimization.

**Loss Function.** Combining *alignment* and *distinctiveness*, we propose a joint optimization objective. The total loss of HAPDA-Finetune is defined as:

$$\mathcal{L} = \mathcal{L}_{\text{ali}} + \lambda \cdot \mathcal{L}_{\text{dis}},$$

where $\lambda$ is a hyperparameter controlling the trade-off between preference alignment and model distinctiveness.

### 3.3 HAPDA-CALIBRATION

The goal of HAPDA-Calibration is to derive a reliable detection score by reweighting each token in the input sequence according to the disagreement between the two specialized models $\mathcal{M}_{\text{hum}}$ and $\mathcal{M}_{\text{mac}}$ obtained from HAPDA-Finetune. This process emphasizes tokens where human and machine preferences diverge and down-weights uncertain predictions.

**Step 1: Measuring Token-wise Prediction Disagreement.** For each token $x_i$ in the input sequence $X = \{x_1, x_2, \ldots, x_n\}$, where $x_{<i}$ denotes the sequence of tokens preceding $x_i$, we first compute the absolute difference between the predicted probabilities from the two models:

$$\Gamma_i = |\pi_{\text{hum}}(x_i \mid x_{<i}) - \pi_{\text{mac}}(x_i \mid x_{<i})|,$$

which captures the degree of disagreement between human and machine. A larger $\Gamma_i$ indicates greater divergence and conveys stronger discriminative power.

**Step 2: Adjusting for Model Uncertainty.** We introduce an entropy-based regularization term to avoid placing high importance on predictions made under uncertainty. Specifically, we compute the entropy of the two predictive distributions at position $x_i$, for $d \in \{\text{hum}, \text{mac}\}$:

$$E_d(x_i) = - \sum_{x \in V} \pi_d(x \mid x_{<i}) \log \pi_d(x \mid x_{<i}).$$

Then the uncertainty is expressed as:

$$U_i = \frac{1}{2 \log V} (E_{\text{hum}}(x_i) + E_{\text{mac}}(x_i)),$$

where $V$ is the vocabulary size.[1] The quantity $1 - U_i$, where $U_i \in [0, 1]$, serves as a confidence score for the disagreement.

**Step 3: Computing Token Weights and Final Score.** Combining the prediction disagreement and confidence adjustment, we first compute the unnormalized weight for each token:

$$\tilde{w}_i = \Gamma_i \cdot (1 - U_i).$$

Note that $\tilde{w}_i$ lies within the interval $[0, 1]$ and is positively correlated with disagreement while being inversely correlated with uncertainty. We then normalize the weights across the sequence to ensure interpretability and stability: $w_i = \tilde{w}_i / \sum_{j=1}^{n} \tilde{w}_j$.

We take the Likelihood method (Solaiman et al., 2019) as an example to further illustrate the process of adapting HAPDA to downstream MeanZero-based methods. Suppose it employs a proxy model $\mathcal{M}_{\text{pxy}}$, the "AI-likeness" score for each token is computed as: $s_i = \log \pi_{\text{pxy}}(x_i \mid x_{<i})$.[2] Under the "Likelihood+HAPDA" setting, the final detection score of a given text $X$ is calculated as:

$$S_X^* = \sum_{i=1}^{n} \frac{\tilde{w}_i}{\sum_{j=1}^{n} \tilde{w}_j} \cdot \log \pi_{\text{pxy}}(x_i \mid x_{<i})$$

Based on the above discussion, we present the full pseudocode of HAPDA in Appendix F.

# 4 EXPERIMENTS AND MAIN RESULTS

## 4.1 EXPERIMENT SETTINGS

### 4.1.1 SOURCE AND PROXY MODELS

Following the setup in (Xu et al., 2024a), we select nine famous open-source models, including OPT (Zhang et al., 2022b) and Llama3 (Grattafiori et al., 2024), with parameter sizes ranging from 1.5B to 13B, as well as two of the latest closed-source models: ChatGPT and GPT-4 (Hurst et al., 2024), as the source models. In line with prior works such as (Mitchell et al., 2023; Xu et al., 2024a), we adopt GPT-J (Wang & Komatsuzaki, 2021) as the proxy model $\mathcal{M}_{\text{pxy}}$ for all methods unless otherwise specified. A more detailed introduction to the models can be found in Appendix A.

### 4.1.2 DATASETS

We conduct the evaluation experiments on four topic-specific subsets: **Books**, **Reviews**, **Wiki**, and **News**, from the AGTD benchmark RAID (Dugan et al., 2024). Each subset contains 150 human-written examples, each containing 150 to 300 words. For each human-written example $H$, we use the text from the `prompt` field $P$ as input to the source models to generate an LLM-produced continuation $M$. For fine-tuning, we select four additional subsets from RAID: **Abstract**, **Poetry**, **Recipes**, and **Reddit**. The generation models differ from the source and proxy models, including eight LLMs such as MPT (Team, 2023) and Cohere (Alnumay et al., 2025). During the construction of each subset, for each LLM, we randomly select 160 unique human-written examples and pair them with their corresponding LLM-generated texts to form human-machine text pairs. Noted that all texts in the fine-tuning dataset are provided by RAID and do not require additional generation. More details on the dataset can be found in Appendix A.

---

[1] We assume that $\mathcal{M}_{\text{hum}}$ and $\mathcal{M}_{\text{mac}}$ originate from the same base model to ensure that $V$ is consistent.

[2] Additional expressions for more MeanZero methods are provided in Appendix B.

Table 1: Overall detection results under the white-box scenario. All methods use the source model for scoring. The AUROC values for each method are averaged across Books, Reviews, Wiki, and News (same for Tables 2 and 3, as well as Figures 4-8).

| Methods/Models | GPT-2 | Neo-2.7 | OPT-2.7 | GPT-J | BLOOM-7 | Falcon-7 | Llama3-8 | OPT-13 | Llama2-13 | Avg. |
|---|---|---|---|---|---|---|---|---|---|---|
| Entropy | 54.1 | 53.4 | 49.6 | 55.4 | 62.2 | 58.6 | 27.8 | 52.7 | 58.9 | 52.5 |
| **+HAPDA** | **64.5** | **63.3** | **65.0** | **65.9** | **73.5** | **72.3** | **49.1** | **64.8** | **65.1** | **64.8** |
| LogRank | 91.7 | 82.5 | 80.3 | 74.2 | 82.5 | 67.7 | 62.3 | 73.6 | 59.8 | 75.0 |
| **+HAPDA** | **93.9** | **88.5** | **83.3** | **79.6** | **87.8** | **71.5** | **69.0** | **80.6** | **68.4** | **80.3** |
| Likelihood | 94.5 | 87.5 | 85.8 | 80.1 | 87.2 | 73.0 | 93.8 | 78.9 | 61.7 | 82.5 |
| DetectLRR | 92.3 | 86.7 | 88.2 | 82.6 | 85.9 | 76.9 | **99.0** | 84.1 | 70.0 | 85.1 |
| **+HAPDA** | **94.3** | **88.4** | **90.0** | **88.8** | **88.3** | **80.3** | 98.7 | **87.0** | **77.1** | **88.1** |
| DetectGPT | 95.0 | 91.2 | 91.9 | 87.5 | 90.8 | 82.3 | **99.4** | 88.6 | 74.7 | 89.0 |
| DetectNPR | 97.0 | 95.7 | 94.8 | 93.1 | 95.6 | 89.6 | 99.2 | 92.4 | 82.6 | 93.3 |
| **+HAPDA** | **97.7** | **96.9** | **95.6** | **95.7** | **96.8** | **93.2** | 99.3 | **94.3** | **88.7** | **95.4** |
| Fast-DetectGPT | 99.7 | 99.0 | 98.6 | 98.3 | **99.6** | 97.9 | **99.9** | 98.3 | 94.9 | 98.5 |
| **+HAPDA** | **99.8** | **99.2** | **99.0** | **98.7** | 99.5 | **98.6** | 99.8 | **98.7** | **95.8** | **98.8** |

### 4.1.3 FINE-TUNING DETAILS

We also adopt GPT-J as the reference model $\mathcal{M}_{ref}$, and to reduce training costs, we use its 4-bit quantized version (Dettmers et al., 2022). Inspired by (Hao et al., 2025), we apply the LoRA technique (Hu et al., 2022), with a rank of 16 and a scaling factor of 32. The adaptation modules include `q_proj` and `v_proj`. The loss function uses a default hyperparameter $\lambda = 0.3$. Following the common hyperparameter configurations, we set the batch size to 16, the optimizer to `AdamW` (Loshchilov & Hutter, 2017), the learning rate to 5e-5, the number of epochs to 10, and apply early stopping with a patience of 2. All experiments are conducted on two NVIDIA A100 80GB GPUs.

### 4.1.4 BASELINES AND METRICS

We select five MeanZero-based methods that are compatible with our task definition as baselines: Entropy (Gehrmann et al., 2019), LogRank (Solaiman et al., 2019), Likelihood (Solaiman et al., 2019), DetectGPT (Mitchell et al., 2023), and Fast-DetectGPT (Bao et al., 2024). In addition, we compare our method with two closely related approaches: DetectLRR and DetectNPR (Su et al., 2023). Detailed descriptions of all methods can be found in Appendix B. Following prior works (Mitchell et al., 2023; Xu et al., 2024a), we adopt AUROC as the binary classification metric for the AGTD task.

### 4.2 MAIN RESULTS

### 4.2.1 HAPDA-FINETUNE RESULTS

As shown in Figure 3(a), the probability distribution of AI-generated texts obtained from the proxy model is generally higher than that of human-written texts, which is consistent with findings in prior studies (Mitchell et al., 2023; Mao et al., 2024). As illustrated in Figure 3(b), the probability gap between AI-generated texts and human-written texts under $\mathcal{M}_{mac}$, obtained via HAPDA fine-tuning, is larger than that observed in Figure 3(a). This indicates that $\mathcal{M}_{mac}$ exhibits a stronger preference for AI-generated texts. Similarly, as shown in Figure 3(c), $\mathcal{M}_{hum}$ shows a stronger preference for human-written texts. Therefore, both $\mathcal{M}_{mac}$ and $\mathcal{M}_{hum}$ exhibit consistent preferences across texts from different origins, which provides a reliable foundation for the subsequent calibration process.

### 4.2.2 WHITE-BOX DETECTION RESULTS

In the white-box setting, the proxy model is identical to the source model. As shown in Table 1, applying HAPDA leads to performance improvements for most source models across the MeanZero baselines under the white-box scenario. Specifically, with the integration of HAPDA, the average AUROC scores of Entropy, LogRank, Likelihood, DetectGPT, and Fast-DetectGPT increase by

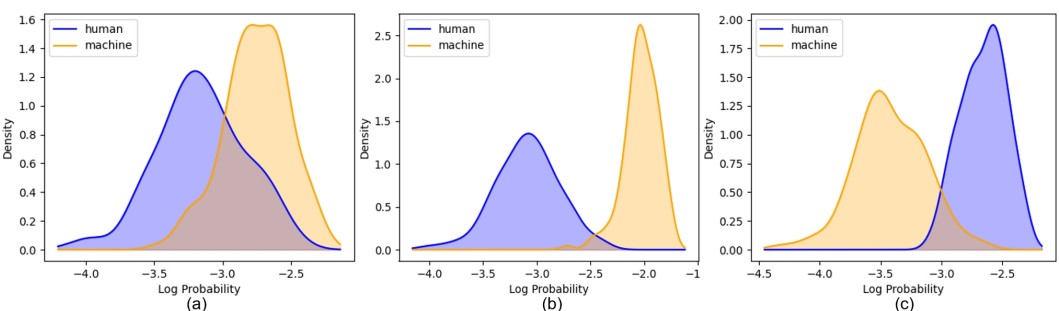

Figure 3: Probability distributions of test documents under $\mathcal{M}_{ref}$ (a), $\mathcal{M}_{mac}$ (b), and $\mathcal{M}_{hum}$ (c) in the Books/GPT-4 setting.

Table 2: Overall detection results under the black-box scenario.

| Methods/Models | GPT-2 | Neo-2.7 | OPT-2.7 | BLOOM-7 | Falcon-7 | Llama3-8 | OPT-13 | Llama2-13 | ChatGPT | GPT-4 | Avg. |
|---|---|---|---|---|---|---|---|---|---|---|---|
| Entropy | 39.6 | 34.7 | 36.5 | 35.9 | 46.5 | 28.4 | 42.9 | 41.0 | 44.8 | 40.2 | 39.1 |
| **+HAPDA** | **46.1** | **48.9** | **53.2** | **46.6** | **57.4** | **48.7** | **57.6** | **47.4** | **62.3** | **46.7** | **51.5** |
| LogRank | 36.7 | 36.4 | 43.7 | 33.1 | 49.6 | 69.2 | 54.6 | 49.5 | 50.2 | 47.1 | 47.0 |
| **+HAPDA** | **44.5** | **49.1** | **50.8** | **43.4** | **59.4** | **73.3** | **61.1** | **57.2** | **65.5** | **55.6** | **56.0** |
| Likelihood | 42.9 | 43.8 | 50.4 | 39.3 | 53.7 | 90.6 | 59.8 | 54.3 | 55.1 | 52.6 | 54.3 |
| DetectLRR | 54.1 | 58.8 | 66.7 | 53.2 | 62.3 | 98.2 | 71.8 | 61.4 | 63.1 | 61.2 | 65.1 |
| **+HAPDA** | **69.0** | **71.1** | **78.7** | **62.5** | **74.9** | **98.9** | **76.8** | **67.3** | **73.5** | **73.7** | **74.6** |
| DetectGPT | 60.1 | 65.1 | 72.0 | 60.1 | 68.6 | **98.8** | 76.8 | 65.5 | 67.4 | 66.1 | 70.1 |
| DetectNPR | 73.2 | 78.1 | 81.3 | 76.5 | 80.7 | 97.8 | 84.2 | 73.9 | 76.4 | 74.1 | 79.6 |
| **+HAPDA** | **83.6** | **81.6** | **85.1** | **83.1** | **87.0** | 98.6 | **88.1** | **77.5** | **79.7** | **81.1** | **84.5** |
| Fast-DetectGPT | 83.1 | 83.7 | 83.6 | 77.4 | 79.7 | 99.2 | 89.1 | 72.1 | 76.9 | 75.2 | 82.0 |
| **+HAPDA** | **88.5** | **85.6** | **86.9** | **85.9** | **87.1** | **99.4** | **92.4** | **83.0** | **85.5** | **82.9** | **87.7** |

12.3%, 5.3%, 5.6%, 6.4%, and 0.3%, respectively. Moreover, compared to the correction-based methods DetectLRR and DetectNPR, HAPDA achieves a higher average AUROC by 3.0% and 2.1%, respectively.

### 4.2.3 BLACK-BOX DETECTION RESULTS

In the black-box setting, the proxy model differs from the source model. As shown in Table 2, applying HAPDA in the black-box scenario leads to significant performance improvements for most source models across the MeanZero baselines. Specifically, with the incorporation of HAPDA, the average AUROC scores of Entropy, LogRank, Likelihood, DetectGPT, and Fast-DetectGPT increase by 12.4%, 9.0%, 20.3%, 14.4%, and 5.0%, respectively. In addition, compared to the correction-based methods DetectLRR and DetectNPR, HAPDA achieves a higher average AUROC by 9.5% and 4.9%, respectively. The comparison between the MeanZero method enhanced with HAPDA and latest zero-shot detectors is discussed in Appendix E. The black-box setting represents a more realistic and widely applicable scenario than the white-box setting. Therefore, all subsequent experiments in this work are conducted by default under the black-box setting.

### 4.2.4 HYPERPARAMETERS SENSITIVITY ANALYSIS

In our work, $\lambda$ is the trade-off coefficient between the alignment loss and the distinctiveness loss. A larger value of $\lambda$ indicates that the model focuses more on distinctiveness, while a smaller value emphasizes preference alignment. As demonstrated by the experimental results in Figure 4, when $\lambda = 0.3$, the HAPDA-enhanced methods—Entropy, Likelihood, LogRank, and FastDetectGPT—achieve the best performance. A larger $\lambda$ leads to a degradation in detection performance. Therefore, we recommend setting $\lambda = 0.3$ as a reasonable hyperparameter configuration.

Table 3: Ablation study for HAPDA. The AUROC values for each method are averaged across all ten source models in the black-box setting. "-D" denotes removing *Distinctiveness* ($\lambda = 0$) during the HAPDA-Finetune process. "-U" denotes removing *Uncertainty* ($U_i = 0$) during the HAPDA-Calibration process.

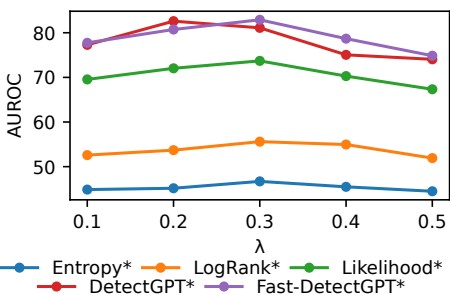

Figure 4: Hyperparameters sensitivity analysis results under five $\lambda$ values: 0.1, 0.2, 0.3, 0.4, 0.5, with GPT-4 as the source model. "*" denotes "+HAPDA".

| Methods/Settings | Ours | -D | -U | -D-U |
|---|---|---|---|---|
| Entropy+HAPDA | **51.5** | 47.9 | 48.6 | 42.1 |
| LogRank+HAPDA | **56.0** | 53.6 | 53.3 | 50.2 |
| Likelihood+HAPDA | **74.6** | 71.8 | 72.0 | 68.4 |
| DetectGPT+HAPDA | **84.5** | 82.2 | 82.5 | 81.1 |
| Fast-DetectGPT+HAPDA | **87.7** | 85.7 | 85.9 | 84.1 |

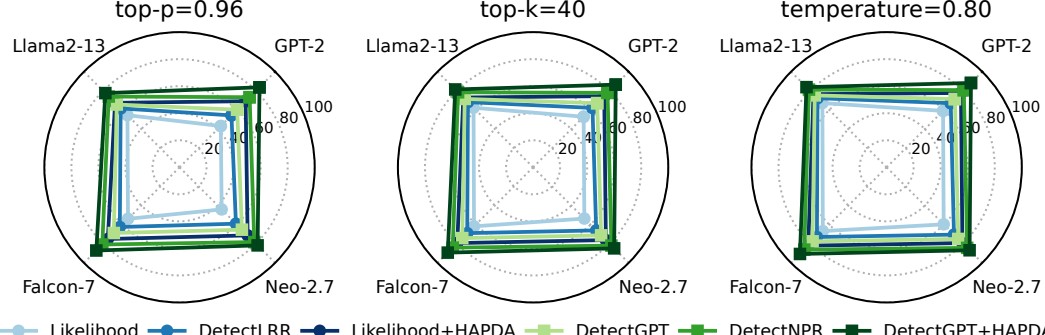

Figure 5: Detection results under different decoding strategies: "top-p=0.96", "tok-k=40" and "temperature=0.80". The source models are GPT-2, Neo-2.7, Falcon-7, and Llama2-13.

### 4.2.5 ABLATION STUDY

Our ablation study focuses on two components: *Distinctiveness* in the fine-tuning stage and *Uncertainty* in the calibration stage. As shown in Table 3, compared to the base variant (-D-U), adding *Distinctiveness* during fine-tuning (-U) improves AUROC by 1.4%–6.5%, while adding *Uncertainty* during calibration (-D) brings a gain of 1.1%–5.8%. The complete HAPDA strategy achieves the best performance, improving AUROC by 3.4%–9.4% across all five base methods.

### 4.3 ROBUSTNESS ANALYSIS

### 4.3.1 DECODING STRATEGIES

Following the setup of (Xu et al., 2024a), we re-implement our experiments under different decoding strategies. Specifically, top-p, top-k, and temperature control the generation quality of the source model from various perspectives. As shown in Figure 5, under all three decoding strategies, HAPDA consistently and significantly improves the detection performance of the vanilla MeanZero method, and also outperforms DetectLRR and DetectNPR.

### 4.3.2 PROXY MODELS SELECTION

In addition to GPT-J, we investigate the detection performance under other proxy models. As demonstrated by the experimental results in Figure 6, HAPDA consistently and significantly improves the detection performance of MeanZero baselines across all proxy models. Specifically, compared to vanilla Likelihood and DetectLRR, HAPDA-enhanced Likelihood achieves AUROC gains ranging from 15.9% to 21.2% and 9.2% to 14.8%, respectively. Compared to vanilla DetectGPT and DetectNPR, HAPDA-enhanced DetectGPT yields AUROC improvements of 7.8% to 18.5% and 3.4% to 9.5%, respectively.

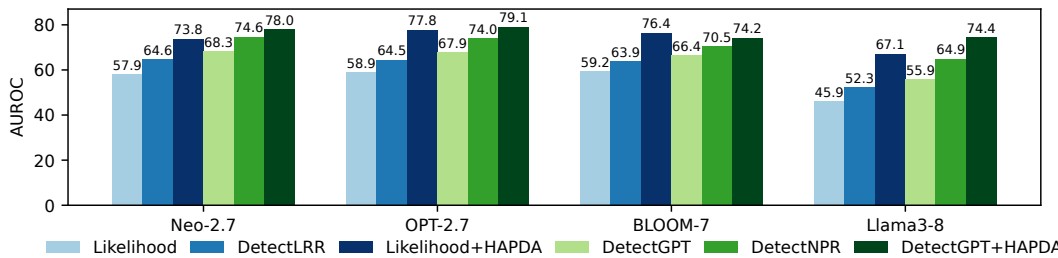

Figure 6: Detection results under four proxy models: Neo-2.7, OPT-2.7, BLOOM-7, Llama3-8, with GPT-4 as the source model.

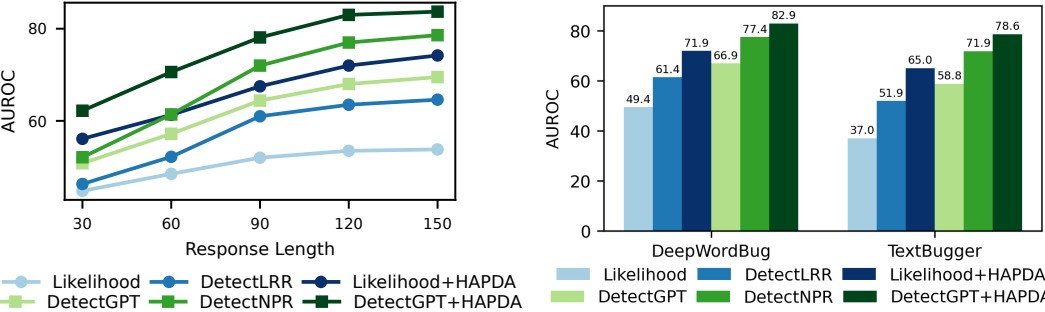

Figure 7: Average AUROC for all ten source models in black-box setting under five response lengths: 30, 60, 90, 120, 150.

Figure 8: Average AUROC for all ten source models in black-box setting under two adversarial attacks: DeepWordBug and TextBugger.

### 4.3.3 RESPONSE LENGTHS

As noted by Mao et al. (2024), the input length can affect the AGTD methods' performance. We truncate human-written and corresponding AI-generated texts to different lengths to re-implement our experiments. As demonstrated by the experimental results in Figure 7, HAPDA consistently improves the detection performance of the MeanZero method even on short texts. Notably, when the text length is relatively short (e.g., response length of 30 or 60), the performance gain of HAPDA is more pronounced compared to DetectLRR and DetectNPR.

### 4.3.4 ADVERSARIAL ATTACKS

We adopt two challenging adversarial attack strategies from TextAttack (Morris et al., 2020): Deep-WordBug (Gao et al., 2018) and TextBugger (Li et al., 2019) to evaluate the robustness of HAPDA under perturbations. Our implementation follows the settings described in (Morris et al., 2020). We provide a detailed description of them in Appendix C. As demonstrated by the experimental results in Figure 8. Although the detection performance of our proposed MeanZero method degrades slightly compared to the non-adversarial setting, it still achieves the best overall performance. Compared to DetectLRR and DetectNPR, our method outperforms them by 10.5% and 5.5% under DeepWord-Bug, and by 13.1% and 6.7% under TextBugger.

## 5 CONCLUSION

In our work, we propose a novel adapter named HAPDA from the perspective of human-machine prediction discrepancy. It is designed to enhance the detection performance of the MeanZero detectors. We introduce a new joint fine-tuning strategy to obtain high-quality human/machine preference models, which are then used to calibrate the downstream detection scores. Extensive experiments demonstrate that across various detection scenarios (white-box, black-box, adversarial attacks, and short texts) and different experimental settings (proxy models and decoding strategies), the introduction of HAPDA consistently and significantly improves the detection performance of baselines.

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

# A  Details of Datasets

## A.1  LLMs in Our Work

In our work, the evaluation dataset consists of texts sourced from: GPT-2 (Radford et al., 2019), Neo-2.7 (Black et al., 2021), OPT-2.7 (Zhang et al., 2022a), GPT-J (Wang & Komatsuzaki, 2021), BLOOM-7 (Workshop et al., 2022), Falcon-7 (Penedo et al., 2023), Llama3-8 (Grattafiori et al., 2024), OPT-13 (Zhang et al., 2022a), Llama2-13 (Touvron et al., 2023), ChatGPT (Hurst et al., 2024), GPT-4o (Hurst et al., 2024). The fine-tuning dataset consists of texts sourced from: Mistral (Jiang et al., 2023), Mistral-Chat (Jiang et al., 2023), MPT (Team, 2023), MPT-Chat (Team, 2023), LLaMA-Chat (Touvron et al., 2023), GPT-3 (Brown et al., 2020), Cohere (Alnumay et al., 2025), Cohere-Chat (Alnumay et al., 2025). The names, parameter sizes, and model versions of all LLMs involved in this paper are listed in Table 4.

Table 4: Details of the models involved in our work.

| Application | Model | Version | Parameters |
|---|---|---|---|
| Evaluation | GPT-2 | openai-community/gpt2-xl | 1.5B |
|  | Neo-2.7 | EleutherAI/gpt-neo-2.7B | 2.7B |
|  | OPT-2.7 | facebook/opt-2.7b | 2.7B |
|  | GPT-J | EleutherAI/gpt-j-6B | 6B |
|  | BLOOM-7 | bigscience/bloom-7b1 | 7B |
|  | Falcon-7 | tiiuae/falcon-7b | 7B |
|  | Llama3-8 | meta-llama/Llama-3.1-8B | 8B |
|  | OPT-13 | facebook/opt-13b | 13B |
|  | Llama2-13 | TheBloke/Llama-2-13B-fp16 | 13B |
|  | ChatGPT | gpt-3.5-turbo-0613 | N.A. |
|  | GPT-4 | gpt-4-0613 | N.A. |
| Fine-tune | Mistral | mistralai/Mistral-7B-v0.1 | 7B |
|  | Mistral-Chat | mistralai/Mistral-7B-Instruct-v0.1 | 7B |
|  | MPT | mosaicml/mpt-30b | 30B |
|  | MPT-Chat | mosaicml/mpt-30b-chat | 30B |
|  | LLaMA-Chat | meta-llama/Llama-2-70b-chat-hf | 70B |
|  | GPT-3 | text-davinci-002 | 175B |
|  | Cohere | command (co.generate()) | N.A. |
|  | Cohere-Chat | command (co.chat()) | N.A. |

## A.2  Evalution Datasets

We select four independent and representative subsets from RAID—Books, Reviews, Wiki, and News—to evaluate both black-box and white-box scenarios. They respectively contain plot-centric summaries of books along with their titles, movie reviews from IMDb along with the names of the movies, introductions to various Wikipedia articles, and BBC news articles with associated titles. For each subset, we randomly sample 150 human-written texts. For "ChatGPT" and "GPT-4," the corresponding machine-generated texts are directly provided by RAID. For the other models, we use the `prompt` field associated with each human-written sample from RAID to generate continuations as machine-generated texts. Consequently, each complete subset consists of 150 negative (human-written) and 150 positive (LLM-generated) samples.

## A.3  Fine-tuning Datasets

For the selection of the fine-tuning training set, we deliberately chose texts that are as unrelated as possible to the topics of the evaluation datasets, in order to verify that HAPDA can achieve good generalization across different text domains. For fine-tuning, we select four additional topic-specific subsets from RAID: Abstract, Poetry, Recipes, and Reddit. They respectively contain abstracts scraped from ArXiv together with paper titles, poems collected from poemhunter.com along with

their titles and genres, recipes with their dish names, and Reddit posts with their titles. The generation models used in this stage differ from the source and proxy models, including eight additional LLMs shown in Table 4. During the construction of each subset, for each LLM, we randomly select 160 unique human-written examples and pair them with their corresponding LLM-generated texts to form human-machine text pairs. Noted that all texts in the finetuning dataset are provided by RAID and do not require additional generation. Consequently, each complete subset consists of 1280 negative and 1280 positive samples.

## B  DETAILS OF BASELINES

We select five MeanZero baselines: Entropy, LogRank, Likelihood, DetectGPT, and Fast-DetectGPT, as well as two methods closely related to our work, DetectNPR and DetectLRR. Unless otherwise specified, their configurations remain consistent with the original papers. We use the **boldface** text to highlight the "AI-likeness" scores $s_i$ used by each method. Here, $\pi_{\text{pxy}}(x_i \mid x_{<i})$ denotes the probability assigned to token $x_i$ by a proxy model $\mathcal{M}_{\text{pxy}}$ given the preceding context $x_{<i}$, and $\text{rank}(x_i \mid x_{<i})$ represents the descending order index of $x_i$ in the predicted vocabulary distribution.

- **Entropy** (Gehrmann et al., 2019) assumes that AI-generated text exhibits lower "openness". The **average entropy** of each token is calculated based on the probability distribution over the vocabulary $\mathcal{V}$:

$$s_i = -\sum_{w \in \mathcal{V}} \pi_{\text{pxy}}(w \mid x_{<i}) \log \pi_{\text{pxy}}(w \mid x_{<i}),$$

  where $\mathcal{V}$ is the vocabulary space.

- **LogRank** (Solaiman et al., 2019) examines the logarithmic ranks of tokens generated by a language model. The **average log rank** of all tokens in the candidate text is used as the metric:

$$s_i = \log \text{rank}_{\text{pxy}}(x_i \mid x_{<i}),$$

  where $\text{rank}_{\text{pxy}}(x_i \mid x_{<i})$ denotes the position of token $x_i$ when sorting all vocabulary tokens in descending order of predicted probability.

- **Likelihood** (Solaiman et al., 2019) assumes that AI-generated text has higher "precision". The **average log probability** of all tokens is used as the metric:

$$s_i = \log \pi_{\text{pxy}}(x_i \mid x_{<i}).$$

- **DetectGPT** (Mitchell et al., 2023) assumes that generated text typically lies in regions of negative curvature in the model's log-probability space. The **perturbation discrepancy** is used as the metric:

$$s_i = \log \pi_{\text{pxy}}(x_i \mid x_{<i}) - \mathbb{E}_{\tilde{x}_i \sim K}\left[\log \pi_{\text{pxy}}(\tilde{x}_i \mid x_{<i})\right],$$

  where $\tilde{x}_i \sim K(\cdot \mid x)$ denotes a perturbed context generated by applying small, meaning-preserving modifications to $x_{<i}$ using a perturbation function $K$.

- **Fast-DetectGPT** (Bao et al., 2024) improves upon DetectGPT by replacing the perturbation step with a more efficient sampling strategy. The **sampling discrepancy** is used as the metric:

$$s_i = \frac{\log \pi_{\text{pxy}}(x_i \mid x_{<i}) - \mu_\sim}{\sigma_\sim},$$

  where:

$$\mu_\sim = \mathbb{E}_{x_i^{(j)} \sim \mathcal{Q}}\left[\log \pi_{\text{pxy}}(x_i^{(j)} \mid x_{<i})\right], \quad \sigma_\sim^2 = \mathbb{E}_{x_i^{(j)} \sim \mathcal{Q}}\left[\left(\log \pi_{\text{pxy}}(x_i^{(j)} \mid x_{<i}) - \mu_\sim\right)^2\right],$$

  and $x_i^{(j)} \sim \mathcal{Q}$ denotes the $j$-th sampled context generated by the sampling model $\mathcal{Q}$.

- **DetectLRR** (Su et al., 2023) is a revised version of LogRank, designed to incorporate both the confidence and rarity of tokens. It uses the **ratio of log-likelihood to log-rank** as the score:

$$s_i = \frac{\log \pi_{\text{pxy}}(x_i \mid x_{<i})}{\log \text{rank}_{\text{pxy}}(x_i \mid x_{<i})}.$$

- **DetectNPR** (Su et al., 2023) is a revised version of DetectGPT, which measures the stability of token ranking under small perturbations. It uses the **normalized perturbed log-rank** as the metric:

$$s_i = \frac{1}{n} \sum_{p=1}^{n} \frac{\log \operatorname{rank}_{\text{pxy}}(x_i \mid \tilde{x}_{<i}^{(p)})}{\log \operatorname{rank}_{\text{pxy}}(x_i \mid x_{<i})},$$

where $\tilde{x}_{<i}^{(p)}$ is the $p$-th perturbed version of the context $x_{<i}$, and $n$ is the total number of perturbations.

## C   DETAILS OF ADVERSARIAL ATTACKS

We adopt two challenging adversarial attack strategies from TextAttack (Morris et al., 2020): **Deep-WordBug** (Gao et al., 2018) and **TextBugger** (Li et al., 2019) to evaluate the robustness of HAPDA under perturbations. These methods generate adversarial yet fluent texts by applying various transformations. Our implementations follow the configurations described in TextAttack (Morris et al., 2020).

- **DeepWordBug** (Gao et al., 2018) is an untargeted attack for classification tasks based on character-level Levenshtein edit distance constraints. It applies four types of character-level perturbations: *Character Insertion*, *Character Deletion*, *Neighboring Character Swap*, and *Character Substitution*. The search method used is a greedy approach called *Word Importance Ranking* (WIR). In our setup, we adopt the default TextAttack configuration.

- **TextBugger** (Li et al., 2019) is an untargeted black-box attack for classification, which maintains semantic similarity measured by cosine similarity on sentence embeddings. It utilizes similar character-level perturbations as DeepWordBug (*Insertion*, *Deletion*, *Swap*, *Substitution*). The search also uses a greedy Word Importance Ranking method. Our implementation follows the default TextAttack settings.

## D   THEORETICAL JUSTIFICATION VIA MUTUAL INFORMATION

Our key intuition is that tokens exhibiting larger divergence between human and machine prediction distributions carry higher mutual information regarding the author identity, thus being more discriminative than treating all tokens uniformly. In this section, we theoretically validate the feasibility of our intuition from the perspective of mutual information.

Formally, let the author identity $A$ be a binary random variable where $A = 0$ represents a human author and $A = 1$ a machine author, with equal prior probabilities $P(A = 0) = P(A = 1) = \frac{1}{2}$.

For a token $x_i$ at position $i$, we define its divergence $\delta_i$ as the total variation distance between the human prediction distribution $P_h(x_i \mid \text{context})$ and the machine prediction distribution $P_m(x_i \mid \text{context})$ over the vocabulary $\mathcal{V}$:

$$\delta_i = \frac{1}{2} \sum_{x \in \mathcal{V}} |P_h(x_i = x \mid \text{context}) - P_m(x_i = x \mid \text{context})|,$$

where by definition $\delta_i \in [0, 1]$.

The mutual information (MI) between token $x_i$ and the author identity $A$ quantifies the reduction in uncertainty about $A$ given the token $x_i$, and is expressed as:

$$I(A; x_i) = H(A) - H(A \mid x_i),$$

where $H(A)$ is the entropy of the author identity, and $H(A \mid x_i)$ is the conditional entropy given the token.

Assuming the token distribution depends on the author's identity, we have:

$$P(x_i \mid A) = \begin{cases} P_h(x_i \mid \text{context}) & \text{if } A = 0, \\ P_m(x_i \mid \text{context}) & \text{if } A = 1. \end{cases}$$

Under this assumption, the mutual information $I(A; x_i)$ corresponds exactly to the Jensen-Shannon divergence (JS) between the two distributions $P_h$ and $P_m$:

$$I(A; x_i) = \text{JS}(P_h \| P_m),$$

where the Jensen-Shannon divergence is defined as

$$\text{JS}(P_h \| P_m) = \frac{1}{2} \left[ \text{KL}(P_h \| M) + \text{KL}(P_m \| M) \right],$$

with $M = \frac{1}{2}(P_h + P_m)$ being the mixture distribution and $\text{KL}(\cdot \| \cdot)$ the Kullback-Leibler divergence.

We now relate the total variation distance $\delta_i$ to the mutual information $I(A; x_i)$. Using the established relationship between JS and total variation distance, we have the following lower bound:

$$\text{JS}(P_h \| P_m) \geq \frac{1}{2} \delta_i^2.$$

From the Pinsker's inequality applied to each KL-divergence term:

$$\text{KL}(P_h \| M) \geq 2(\text{TV}(P_h, M))^2, \quad \text{KL}(P_m \| M) \geq 2(\text{TV}(P_m, M))^2,$$

where $\text{TV}(\cdot, \cdot)$ denotes total variation distance. Since $M = \frac{1}{2}(P_h + P_m)$, we have:

$$\text{TV}(P_h, M) = \text{TV}(P_m, M) = \frac{1}{2} \delta_i.$$

Substituting these into the JS definition:

$$\text{JS}(P_h \| P_m) = \frac{1}{2} \left[ \text{KL}(P_h \| M) + \text{KL}(P_m \| M) \right] \geq \frac{1}{2} \left[ 2 \left( \frac{1}{2} \delta_i \right)^2 + 2 \left( \frac{1}{2} \delta_i \right)^2 \right] = \frac{1}{2} \delta_i^2.$$

Thus we obtain the fundamental relationship:

$$I(A; x_i) \geq \frac{1}{2} \delta_i^2.$$

This bound establishes that tokens with larger divergence $\delta_i$ necessarily have higher mutual information about author identity. Consider two tokens $x_i$ and $x_j$ such that $\delta_i > \delta_j$. It follows that:

$$I(A; x_i) \geq \frac{1}{2} \delta_i^2 > \frac{1}{2} \delta_j^2,$$

meaning $x_i$ provides a strictly higher lower bound on mutual information than $x_j$. Moreover, since JS is a metric and increases with distribution divergence, $\delta_i > \delta_j$ typically implies $I(A; x_i) > I(A; x_j)$ in practice.

From a classification perspective, the minimal error probability $P_e$ for author attribution is bounded by Fano's inequality:

$$H(A \mid \mathbf{x}) \leq H(P_e) + P_e \log_2(|\mathcal{A}| - 1),$$

where $\mathbf{x} = \{x_1, \ldots, x_n\}$ denotes the sequence of tokens. Maximizing the sum of mutual information of selected tokens:

$$\max_{\mathbf{x}} \sum_{i=1}^{n} I(A; x_i),$$

corresponds to minimizing the conditional entropy $H(A \mid \mathbf{w})$ and thus the classification error $P_e$.

In summary, tokens with larger prediction divergence $\delta_i$ exhibit higher mutual information $I(A; x_i)$ about author identity, as both the lower bound $\frac{1}{2} \delta_i^2$ and the actual JS value increase with $\delta_i$. Therefore, prioritizing tokens based on $\delta_i$ leads to maximal information gain and reduces uncertainty in authorship attribution more effectively than uniform weighting of all tokens.

## E    COMPARISON WITH ADVANCED ZERO-SHOT DETECTORS

In Section 3.1, we defined our task as primarily aiming to enhance the existing MeanZero method. To enable a more comprehensive evaluation, beyond the baselines introduced in Section 4.1.4, we further include three latest zero-shot detectors outside the MeanZero framework for comparison: DNA-GPT (Yang et al., 2024), Raidar (Mao et al., 2024), and Lastde (Xu et al., 2024a). Specifically, for DNA-GPT, we use GPT-J to compute BScore, set the number of re-prompting iterations to 10, and fix the truncate rate to 0.5. For Raidar, we adopt the "Invariance" strategy, with GPT-4 as the generation model and the prompt template "Rewrite this for me:". For Lastde, we set the sliding window size to 4 and the number of scales to 10. All other configurations follow the original implementations. We redeploy these methods under the black-box detection setting on the Books dataset, with results summarized in Table 5. The DetectGPT detector incorporated with HAPDA achieves AUROC scores comparable to or even surpassing the best zero-shot detectors across multiple tasks. This demonstrates the potential of HAPDA to elevate relatively weaker classifiers to performance levels close to the latest state-of-the-art detectors.

Table 5: Additional Detection results under the black-box scenario for Books datasets. "value" denotes the second-best AUROC.

| Methods/Models | OPT-13 | Llama2-13 | ChatGPT | GPT-4 |
|---|---|---|---|---|
| DetectGPT | 75.9 | 67.8 | 65.2 | 69.4 |
| DetectGPT+HAPDA | 89.3 | **78.9** | 78.8 | **83.3** |
| DNA-GPT | 85.5 | 73.1 | 76.3 | 75.1 |
| Raidar | 84.0 | 76.5 | 78.6 | 78.4 |
| Lastde | **90.5** | 77.2 | **80.6** | 81.5 |

## F    DETAILS OF HAPDA

In Section 3, we provide a detailed description of HAPDA's workflow. In this section, we summarize the procedures of HAPDA-Finetune and HAPDA-Calibration in the pseudocode shown in Algorithms 1 and 2.

## G    LIMITATIONS

Although our proposed approach has been shown to significantly improve the MeanZero method across various application scenarios, HAPDA still has certain limitations: (1) HAPDA is not a complete detection system but is designed as an extension of the existing MeanZero detector, and it is not directly applicable to some zero-shot and supervised methods. In future work, we will explore its transferability to these settings. (2) The introduction of auxiliary preference models in HAPDA inevitably incurs additional runtime and memory overhead during training and inference. While we mitigate this issue in our work using techniques such as quantization and LoRA, we plan to further optimize the framework through lightweight and parallelized designs.

## H    LLM USAGE STATEMENT

In paper writing, we used LLMs solely for text polishing. LLMs were not employed for retrieval and discovery (e.g., finding related work) and research ideation.

---

**Algorithm 1** HAPDA-Finetune

---

1: **Input:** Reference model $\mathcal{M}_{\text{ref}}$, initial weights for $\mathcal{M}_{\text{hum}}, \mathcal{M}_{\text{mac}}$ (copied from $\mathcal{M}_{\text{ref}}$), dataset $\mathcal{D} = \{(P, H, M)\}$ of (prompt, human, machine) triples, trade-off $\lambda$, learning rate $\eta$, batch size $B$, number of epochs $T$.
2: **Output:** Fine-tuned models $\mathcal{M}_{\text{hum}}, \mathcal{M}_{\text{mac}}$.
3: **for** $epoch = 1 \ldots T$ **do**
4:     **for** each mini-batch $\mathcal{B} \subset \mathcal{D}$ of size $B$ **do**
5:         **for** each $(P, H, M) \in \mathcal{B}$ **in parallel do**
6:             compute token-level (or sequence-level) log-probabilities under $\mathcal{M}_{\text{hum}}, \mathcal{M}_{\text{mac}}, \mathcal{M}_{\text{ref}}$:

$$\log \pi_{\text{hum}}(H \mid P), \ \log \pi_{\text{hum}}(M \mid P), \ \log \pi_{\text{mac}}(H \mid P), \ \log \pi_{\text{mac}}(M \mid P),$$

    and similarly for $\mathcal{M}_{\text{ref}}$.
7:             compute preference margins (per triple):

$$\Delta_{\text{mac}} \leftarrow \log \frac{\pi_{\text{mac}}(M \mid P)}{\pi_{\text{ref}}(M \mid P)} - \log \frac{\pi_{\text{mac}}(H \mid P)}{\pi_{\text{ref}}(H \mid P)},$$

$$\Delta_{\text{hum}} \leftarrow \log \frac{\pi_{\text{hum}}(H \mid P)}{\pi_{\text{ref}}(H \mid P)} - \log \frac{\pi_{\text{hum}}(M \mid P)}{\pi_{\text{ref}}(M \mid P)}.$$

8:         **end for**
9:         compute alignment loss over batch:

$$\mathcal{L}_{\text{ali}} \leftarrow -\frac{1}{|\mathcal{B}|} \sum_{(P,H,M) \in \mathcal{B}} \big( \log \sigma(\Delta_{\text{mac}}) + \log \sigma(\Delta_{\text{hum}}) \big).$$

10:         compute token-level distributions $p_{\text{mac}}^Y, p_{\text{hum}}^Y$ for $Y \in \{H, M\}$ and the JS divergence terms,

$$\mathcal{L}_{\text{dis}} \leftarrow \frac{1}{|\mathcal{B}|} \sum_{(P,H,M) \in \mathcal{B}} \Big( 2 \log 2 - \text{JS}(p_{\text{mac}}^M \parallel p_{\text{hum}}^M) - \text{JS}(p_{\text{mac}}^H \parallel p_{\text{hum}}^H) \Big).$$

11:         total loss: $\mathcal{L} \leftarrow \mathcal{L}_{\text{ali}} + \lambda \cdot \mathcal{L}_{\text{dis}}$.
12:         update $\mathcal{M}_{\text{hum}}, \mathcal{M}_{\text{mac}}$ by gradient descent on $\mathcal{L}$ with lr $\eta$ (optionally apply LoRA/quantization-aware steps).
13:     **end for**
14: **end for**
15: **return** $\mathcal{M}_{\text{hum}}, \mathcal{M}_{\text{mac}}$

---

---

**Algorithm 2** HAPDA-Calibration

---

1: **Input:** Fine-tuned preference models $\mathcal{M}_{\text{hum}}, \mathcal{M}_{\text{mac}}$, proxy detector model $\mathcal{M}_{\text{pxy}}$ (used to compute per-token AI-likeness scores), input token sequence $X = (x_1, \ldots, x_n)$, vocabulary size $V$.

2: **Output:** Calibrated detection score $S_X^*$.

3: **for** $i = 1$ to $n$ **do**

4:     compute conditional probabilities for token $x_i$:
$$p_{\text{hum}}(\cdot \mid x_{<i}) \leftarrow \pi_{\text{hum}}(\cdot \mid x_{<i}), \quad p_{\text{mac}}(\cdot \mid x_{<i}) \leftarrow \pi_{\text{mac}}(\cdot \mid x_{<i}).$$

5:     compute disagreement magnitude:
$$\Gamma_i \leftarrow \left| \pi_{\text{hum}}(x_i \mid x_{<i}) - \pi_{\text{mac}}(x_i \mid x_{<i}) \right|.$$

6:     compute entropies:
$$E_{\text{hum}}(x_i) \leftarrow - \sum_{v \in V} p_{\text{hum}}(v \mid x_{<i}) \log p_{\text{hum}}(v \mid x_{<i}),$$
$$E_{\text{mac}}(x_i) \leftarrow - \sum_{v \in V} p_{\text{mac}}(v \mid x_{<i}) \log p_{\text{mac}}(v \mid x_{<i}).$$

7:     compute normalized uncertainty:
$$U_i \leftarrow \frac{1}{2 \log V} \big( E_{\text{hum}}(x_i) + E_{\text{mac}}(x_i) \big).$$

8:     unnormalized weight:
$$\tilde{w}_i \leftarrow \Gamma_i \cdot (1 - U_i).$$

9:     compute per-token AI-likeness score (example: log-prob from proxy model):
$$s_i \leftarrow \log \pi_{\text{pxy}}(x_i \mid x_{<i}).$$

10: **end for**

11: normalize weights: $w_i \leftarrow \tilde{w}_i / \sum_{j=1}^{n} \tilde{w}_j$ (if denominator is 0, fall back to uniform weights).

12: compute final calibrated detection score:
$$S_X^* \leftarrow \sum_{i=1}^{n} w_i \cdot s_i.$$

13: **return** $S_X^*$

---

