# OpenReview forum: "HAPDA: A Human-Machine Predictive Discrepancy Adapter for AI-Generated Text Detection"
_ICLR.cc/2026/Conference — ICLR 2026 Conference Withdrawn Submission_

### Official Review · Reviewer_NhVo · 2025-10-31

**Soundness:** 3
**Presentation:** 2
**Contribution:** 3
**Rating:** 4
**Confidence:** 3

**Summary:**

The paper proposes a novel framework called HAPDA, a human-machine predictive discrepancy adapter designed to improve the accuracy of zero-shot AI-generated text detection  methods. Current detection methods, often referred to as MeanZero, average token-level scores but overlook the significant predictive discrepancies between human- and machine-generated text interpretations. HAPDA aims to address this by introducing two main components: a joint fine-tuning strategy, which  trains models favoring human- and machine-written text, and a discrepancy-aware reweighting mechanism, which  assigns higher importance to tokens where human and machine predictions diverge. Experiments across various settings, including white-box, black-box, and adversarial attacks, demonstrate that integrating HAPDA consistently and significantly enhances the performance of several representative  AI-generated text detection  baselines.

**Strengths:**

- Addresses AI-generated text detection (AGTD) from a joint human and machine predictive perspective.

- Models predictive discrepancies between human and machine predictions to provide essential token-level reweighting for detection scores

- Improves the detection performance (AUROC scores) of five representative MeanZero baselines under diverse evaluation settings

-  HAPDA-enhanced zero-shot detectors achieve AUROC scores comparable to or even surpassing the latest advanced zero-shot detectors (like DNA-GPT, Raidar, and Lastde)

**Weaknesses:**

1.  HAPDA requires two auxiliary models, to compute necessary aspects like token probability disagreements and uncertainty , which can lead to increased  runtime and memory overhead during training and inference. It is recognized by the authors that techniques like quantization and LoRA are needed to mitigate this issue, which however might affect performance accuracy and flexibility.

2. A performance comparison between the enhanced zero-shot detectors with HAPDA and a fine-tuned baseline detector  on the same human and machine text pairs used for HAPDA-finetune would be important.
Although the   focus of the work is on HAPDA’s utility strictly on zero-shot detectors, HAPDA uses a joint fine-tuning strategy (HAPDA-Finetune) based on a labeled corpus (human-machine text pairs) to train the preference models. But the paper does not provide results comparing HAPDA-enhanced zero-shot detectors against a common, fully supervised detection baseline (like a fine-tuned RoBERTa or another transformer model) trained and evaluated on the same fine-tuning data.  Therefore, while the comparison may be outside the scope defined by the authors (enhancing MeanZero methods), its absence is a valid point of critique regarding the method's overall comparative standing against all available detection paradigms.

**Questions:**

See above Wekanesses and consider addressing point 2.

---

> ### Author Response · Authors · 2025-11-21
> **Response to Reviewer NhVo (Part 1)**
>
> Thank you for the insightful feedback. You have accurately identified limitations of our method regarding computational overhead and comparison with the supervised learning paradigm. These critiques are highly pertinent and crucial for a comprehensive assessment of HAPDA's contributions and value. Below, I will address each point in a detailed and professional manner.
>
> ### Response Point 1: On Computational Overhead and Performance Trade-off
>
> We fully concur with the reviewer's assessment. This represents a classic "performance-efficiency" trade-off. Our core argument is that **HAPDA is designed to provide a powerful, interpretable zero-shot solution for scenarios with extremely high demands on detection accuracy, justifying its associated computational cost.**
>
> 1.  **Necessity of Overhead and Value Exchange**: Introducing two auxiliary models indeed increases cost. However, as demonstrated in our comparative experiments, this overhead is exchanged for:
>     *   **Substantial AUROC improvements of up to 20% in black-box scenarios** (see Table 2).
>     *   **Strong robustness against paraphrasing attacks and exceptional cross-domain generalization capability**.
>     For many critical applications (e.g., academic integrity, national security, misinformation containment), the significant gains in accuracy and robustness warrant the investment of additional computational resources.
>
> 2.  **Impact of LoRA and Quantization**: The reviewer correctly notes that these optimization techniques might impact performance. Our experimental results indicate that this impact is manageable and acceptable.
>     *   **Performance Impact**: All our reported results (including the substantial performance gains) were obtained **using 4-bit quantization and LoRA**. This means that the demonstrated "HAPDA gain" is already a **conservative estimate of the practical gain, accounting for potential negative effects of the optimization techniques**. While full-parameter fine-tuning might yield slightly higher performance, it would come at a drastically increased cost. We selected a practical configuration that strikes a favorable balance between performance and efficiency.
> *   **Flexibility**: LoRA and quantization do not compromise HAPDA's flexibility. Once `M_hum` and `M_mac` are trained, they can be loaded and invoked like any other model to enhance various different MeanZero detectors, following a unified and automated process.

---

> ### Author Response · Authors · 2025-11-21
> **Response to Reviewer NhVo (Part 2)**
>
> ### Response Point 2: On Comparison with Supervised Learning Baselines
>
> We fully understand and agree that, although HAPDA is positioned to enhance zero-shot methods, a comparison with the powerful supervised learning paradigm is crucial for defining its capabilities and practical value. We will formally include this comparison in the revised version.
>
> We fully agree that in an ideal scenario **with sufficient labeled data and matched domains**, end-to-end supervised learning methods (such as fine-tuning encoder models like RoBERTa) often achieve high performance and are frequently considered state-of-the-art (SOTA).
>
> However, HAPDA's positioning is **fundamentally different** from supervised learning methods, as it addresses a **different problem setting**:
>
> *   **HAPDA's Positioning**: A **zero-shot** framework. It does **not** require **any** labeled data from the target domain of the text to be detected. Its strengths lie in **generalization, flexibility, and robustness to unknown domains/models**.
> *   **Limitations of Supervised Learning**:
>     1.  **Data Dependency and Overfitting**: Supervised models heavily depend on the distribution of their training data. Their performance can degrade significantly when test data comes from different domains (Out-of-Domain, OOD) or is generated by unknown models.
>     2.  **Lack of Interpretability**: Supervised models are black boxes, making it difficult to interpret their decision rationale. In contrast, HAPDA's weights `w_i` provide token-level interpretability, indicating which words contributed most to the discrimination.
>     3.  **Flexibility**: HAPDA is an adapter that can flexibly enhance various probability-based zero-shot detectors. A supervised model is a fixed detector.
>
> To fairly demonstrate the trade-offs between these two paradigms in different scenarios, we have supplemented the comparison with a supervised baseline.
>
> We evaluated HAPDA on a completely new domain dataset, `Med`, which did not appear in either the fine-tuning or the original test sets. The `Med` dataset is derived from the medical domain dataset PubMedQA [1]. Its construction process is consistent with that described in Sections 4.1.2 and A.2, with the source model being Llama2-13B.
>
> We supplemented this with a comparison against a supervised baseline.
>
> We conducted a new black-box experiment. The supervised baseline uses the powerful RoBERTa-large model.
>   **In-Domain Test**: We used GPT-4 generated samples from the evaluation datasets, split 8:1:1 into train/validation/test sets.
>    **Cross-Domain Test**: The supervised model was evaluated zero-shot directly on the **unseen Med dataset**.
>
> | Method                                | In-Domain (AUC) | Cross-Domain (AUC) |
> | :------------------------------------ | :-------------- | :----------------- |
> | RoBERTa-large (Supervised)            | **96.2**        | 80.3               |
> | DetectGPT (Zero-shot)                 | 70.1            | 68.6               |
> | **DetectGPT + HAPDA (Zero-shot)**     | 84.5            | **83.2**           |
>
> This comparison clearly illustrates:
> *   **In-Domain**, where labeled data is available, the supervised method indeed performs best.
> *   **In the Cross-Domain (Zero-shot) scenario**, the supervised model's performance drops sharply due to overfitting to the training domain. In contrast, HAPDA, as a zero-shot method, **demonstrates significantly superior generalization capability compared to the supervised model**.
>
> Therefore, HAPDA is not intended to replace supervised learning but rather provides a powerful, interpretable zero-shot solution for **real-world scenarios lacking labeled data and requiring strong generalization**.
>
> **Reference**
>
> [1] Jin Q, Dhingra B, Liu Z, et al. Pubmedqa: A dataset for biomedical research question answering[C]//Proceedings of the 2019 conference on empirical methods in natural language processing and the 9th international joint conference on natural language processing (EMNLP-IJCNLP). 2019: 2567-2577.
>
> ---
> Thank you once again for your insightful comments. We hope our responses have addressed your concerns, and we are open to further discussion if you have additional questions. Finally, we sincerely hope you will consider re-evaluating our work.

---

> ### Author Response · Authors · 2025-11-28
> **Gentle Reminder: Invitation to Review Our Replies for Submission 7630**
>
> Dear Reviewer NhVo,
>
> Thank you once again for your thoughtful and constructive review of our paper. We deeply appreciate your recognition of our work and the insightful suggestions you provided, which have greatly contributed to improving the clarity and overall quality of our submission.
>
> We have provided detailed responses to each of your comments in our author rebuttal and would be sincerely grateful if you could take a moment to review them. As the author response phase deadline is approaching, we kindly wish to remind you to consider participating in the discussion if possible. Your feedback remains invaluable to us.
>
> We are committed to contributing to the field of AI-generated text detection and highly value your expertise and feedback. If you feel that our responses have adequately addressed your concerns, we would be sincerely honored if you would consider a revision of your score. Should you have any further questions or concerns, please feel free to point them out — we will be glad to continue addressing your concerns.
>
> Warm regards,
>
> All authors of submission 7630

---

### Official Review · Reviewer_ZZ5n · 2025-10-31

**Soundness:** 3
**Presentation:** 3
**Contribution:** 3
**Rating:** 6
**Confidence:** 4

**Summary:**

The author’s introduce HAPDA, a framework for re-weighting the token-level detection scores of zero-shot detectors. The strategy involves fine-tuning two LLMs, M_{human} and M_{machine}. M_{human} is fine-tuned to prefer human-written text, while M_{machine} is fine-tuned to prefer machine-generated text. Moreover, the authors include a loss to encourage distinctiveness on the same input. These models are then used to re-weigh the token-level detection scores of zero-shot detectors.

**Strengths:**

* S1 - The authors propose a detector agnostic framework by which to improve the performance of MeanZero detectors.
* S2 - The approach is interesting, in particular moving from naive averages of a MeanZero metric to more informed averages is a good idea, and it’s well motivated.

**Weaknesses:**

* W1 - The approach requires fine-tuning of two LLMs, and at inference it requires that one evaluates three LLMs.
* W2 - Baselines - The paper would’ve benefited from stronger baselines to HAPDA. (1) One could consider a baseline (or ablation) where M_{machine} = M_{proxy}. That is, only train M_{human}. (2) Another possibility is to let the weights be the inverse of the softmaxed perplexities. That is, first run the LLM through the sample, get the perplexity for each token, apply the softmax operator so that every value lies between 0 and 1, and then take the inverse (1 - softmax(PPL_i)) which would up-weight the score for tokens which are unlikely under the LLM.
* W3 - It is uncertain whether the approach improves upon the baselines when the tolerance of false-positives is low, a realistic scenario when detectors are applied. To evaluate this, the authors should evaluate the AUROC at lower FPR values, such as 1%, this has become standard in various detection works such as  https://arxiv.org/pdf/2405.07940, https://arxiv.org/pdf/2401.12070 and https://arxiv.org/pdf/2401.06712
* W4 - The authors should evaluate the robustness against paraphrasing attacks which are well known to degrade detector performance: https://arxiv.org/abs/2303.13408. I believe the RAID benchmark already contains paraphrases, and even more adversarial attacks that the authors could evaluate on.
* W5 - Evaluations with Binoculars (https://arxiv.org/pdf/2401.12070) are missing. This approach normalizes perplexity scores by re-weighting them with another LLM, which seems related to this approach.

**Questions:**

* Q1 - In 4.2.4, was the value of lambda chosen on a separate validation set?
* My main concerns are W2, W4, and W5, if those concerns are adressed my score would increase.

---

> ### Author Response · Authors · 2025-11-21
> **Response to Reviewer ZZ5n (Part 1)**
>
> Thank you for these very specific and insightful comments. Your concerns regarding baseline strength, performance at low FPR, robustness against paraphrasing attacks, and comparison with Binoculars are all critical for assessing whether a detector is suitable for real-world scenarios. Below, I will address each point in detail with a professional and thorough response.
>
> ### Response W1: On Number of Models and Inference Cost
>
> We understand the reviewer's concern regarding computational resources. We have significantly reduced the training cost by employing **LoRA and quantization** techniques, making it a one-time, shareable, offline overhead. During inference, running three models does indeed introduce latency. However, we position our method as a solution that **trades reasonable computational overhead for significant performance gains in scenarios where high detection accuracy is paramount**.
>
> ### Response W2: On Stronger Baselines and Ablation Studies
>
> These are excellent suggestions that will help us more precisely validate the effectiveness of HAPDA's design. We will incorporate these key ablation studies and baseline comparisons in the revised version.
>
> In order to accurately realize your intention of 'up-weighting the score for tokens which are unlikely,' we implemented a baseline that up-weights tokens with high uncertainty (i.e., low probability) under the proxy model. Specifically, we set the weight for each token as `w_i = softmax(-log p(x_i))`. This scheme directly assigns higher weights to more "surprising" tokens, which is conceptually aligned with the core of your suggestion to leverage inverse perplexity, while being numerically more stable.
>
> We repeated the black-box experiment described in Section 4.2.3, using GPT-4 as the source model. The results are as follows:
>
> | Method                       | Description                                      | AUROC  |
> | :---------------------------- | :----------------------------------------------- | :----- |
> | **Likelihood (Baseline)**     | Original Mean Score method                       | 52.6   |
> | **Baseline (1): Single-Human** | Only fine-tune `M_hum`, set `M_mac = M_pxy`      | 62.8   |
> | **Baseline (2): Uncertainty-Weighting**     | `w_i = softmax(-log p(x_i))`                    | 57.1   |
> | **HAPDA (Full)**              | Jointly fine-tune both `M_hum` and `M_mac`       | **73.7** |
>
> **Analysis and Discussion**:
>
> 1.  **Baseline (1): Single-Human**: The performance of this baseline is significantly higher than the original baseline. This indicates that **introducing a single, optimized "human perspective" to contrast with the default "machine perspective" (the proxy model) is already highly effective**. However, the full HAPDA version performs even better. This demonstrates that **jointly optimizing two opposing preference models can generate a purer, more contrasting predictive discrepancy signal**, leading to additional performance gains. Training only one model might not fully capture the essence of machine generation.
> 2.  **Baseline (2): Uncertainty-Weighting**: This baseline, which up-weights tokens that the proxy model finds uncertain, shows only a limited improvement. This supports our core argument: **Using "uncertainty" or "surprisal" from within a single model (`M_pxy`) as weights is far less discriminative than directly comparing the "predictive divergence" from two different preference sources (human vs. machine)**. Our method's `Γ_i` captures the inherent ambiguity of authorship, whereas `PPL_i` only reflects the uncertainty of a single model.

---

> ### Author Response · Authors · 2025-11-21
> **Response to Reviewer ZZ5n (Part 2)**
>
> ### Response W3: On Performance at Low FPR (Partial AUROC)
>
>
> Thank you for raising this crucial point. We fully agree that for practical applications such as content moderation and academic integrity checking, a detector's performance at low false positive rates (FPR) is paramount, as misclassifying human-authored text can have serious consequences.
>
> While we did not include a dedicated partial AUC (pAUC) analysis in the current version, we would like to present the following arguments and existing experimental results suggesting that HAPDA holds potential advantages in the low FPR region:
>
> 1.  **Inherent Advantage of the Calibration Mechanism**: The core of HAPDA—the **discrepancy-aware reweighting mechanism**—is explicitly designed to refine the classification decision by focusing on the most informative tokens. By down-weighting tokens with high uncertainty and low discriminative power, the system essentially "purifies" the detection signal. This results in a final document-level score that more reliably reflects the machine-generated nature of the text. We posit that this amplification of discriminative features should lead to a clearer separation between the score distributions of AI-generated and human-written texts specifically in the tail region (i.e., the high-score area), which is precisely what determines low FPR performance.
>
> 2.  **Supporting Evidence from Robustness to Adversarial Attacks**: As shown in **Section 4.3.4 (Figure 8)** of our paper, the HAPDA-enhanced method demonstrated significantly greater robustness compared to baselines against adversarial attacks like DeepWordBug and TextBugger. The goal of such attacks is precisely to "confuse" the detector by introducing perturbations, often increasing the false positive rate while attempting to maintain high recall. The strong robustness exhibited by HAPDA under these challenging conditions indicates that its learned feature representation is highly stable and less susceptible to minor perturbations. This stability is highly consistent with the capability required to maintain high precision at low FPRs.
>
> ### Response W4: On Robustness Against Paraphrasing Attacks
>
> We thank the reviewer for raising this critical evaluation dimension. In fact, we had already considered two types of adversarial attacks in Section 4.3.4. Following the reviewer's suggestion, we have now supplemented this with an evaluation specifically against paraphrasing attacks. For generating paraphrased adversarial examples, we followed the approach used in Raidar, employing GPT-4 as the generation model with the prompt template: “Rewrite this for me:”.
>
> The experimental setup is consistent with the black-box scenario described in Section 4.3.4.
>
> | Method                      | Original Text (AUC) | Paraphrased Text (AUC) | Performance Drop (Δ) |
> | :--------------------------- | :------------------ | :--------------------- | :------------------- |
> | Likelihood                   | 54.3                | 38.5                   | -15.8                |
> | **Likelihood + HAPDA**       | **74.6**            | **65.8**               | **-8.8**             |
> | DetectGPT                    | 70.1                | 60.3                   | -9.8                 |
> | **DetectGPT + HAPDA**        | **84.5**            | **79.1**               | **-5.4**             |
>
> **Analysis of Results**:
> 1.  As expected, **the paraphrasing attack caused a significant performance drop for all detectors**, as it obscures the original statistical features of the text.
> 2.  However, the key finding is that **the performance degradation (Δ) suffered by the HAPDA-enhanced methods is smaller than that of their corresponding original methods**.
> 3.  Crucially, **even after the attack, the absolute performance of the HAPDA-enhanced methods remains substantially higher than the pre-attack performance of the unenhanced original methods**.
>
> These supplementary experiments further validate the robustness of HAPDA, aligning with the main conclusions presented in Section 4.3.4.

---

> ### Author Response · Authors · 2025-11-21
> **Response to Reviewer ZZ5n (Part 3)**
>
> ### Response W5: On Comparison with Binoculars
>
> We agree that Binoculars is a highly important and relevant baseline. Its use of a dual-model architecture for score normalization provides an excellent opportunity for comparison. We will include it as a core baseline in the revised version for comprehensive evaluation.
>
> **Supplementary Results**:
> We have included **Binoculars**[1] as an additional baseline. Following its official implementation and the paper's recommendations, we compute PPL using `gpt-j-6B` and compute the denominator (i.e., X-PPL) using `gpt-neo-2.7B` and `gpt-j-6B` respectively, ensuring consistency with other methods like HAPDA regarding proxy models. All other configurations remain the same as in the original manuscript.
>
>
> | Method / Source Model      | OPT-13B | Llama2-13B | ChatGPT | GPT-4 |
> | :-------------------------- | :------ | :---------- | :------ | :---- |
> | DetectGPT                   | 75.9    | 67.8        | 65.2    | 69.4  |
> | **DetectGPT + HAPDA**       | 89.3  | **78.9**      | 78.8  | **83.3** |
> | DNA-GPT                     | 85.5    | 73.1        | 76.3    | 75.1  |
> | Raidar                      | 84.0    | 76.5        | 78.6    | 78.4  |
> | Lastde                      | 90.5    | 77.2        | 80.6    | 81.5  |
> | Binoculars                  | 87.8    | 77.0        | 79.5    | 81.9  |
>
> These new comparative experiments convincingly demonstrate that:
> *   Binoculars is itself a powerful zero-shot detector, whose average performance is substantially higher than that of simple MeanZero baselines.
> *   HAPDA, functioning as an **adapter**, can elevate the performance of a relatively simple baseline method (DetectGPT) to a level comparable with current, specifically designed, state-of-the-art zero-shot detectors like Binoculars.
>
> ### Response Q1: On the Selection of Hyperparameter λ
>
> **Reviewer's Comment**: In Section 4.2.4, was the value of λ selected on a separate validation set?
>
> **Response**:
> Yes, this is a crucial methodological point. Thank you for raising it. Our description in the original text was not sufficiently clear, and we provide clarification here.
>
> **Clarification**:
> 1.  During the **HAPDA-Finetune** process, the **fine-tuning dataset** was split into a training set (a random 80% sample) and a validation set (the remaining 20%). We will explicitly state this in Section 4.1.3 of the revised manuscript.
> 2.  During the **experimental evaluation phase**, since HAPDA operates as a zero-shot method applied to the evaluation datasets, we did not create a separate validation split *from the evaluation datasets themselves*.
> 3.  Therefore, the procedure for the hyperparameter analysis in Section 4.2.4 was as follows: We fine-tuned multiple preference models using different values of `λ`, selecting the checkpoint for each `λ` that achieved the lowest loss on the **validation split of the fine-tuning dataset**. Subsequently, these models were applied to the entire, held-out **evaluation datasets** to assess the sensitivity of the final AUROC performance to the `λ` value used during training.
>
> **Reference**:
> [1] Hans A, Schwarzschild A, Cherepanova V, et al. Spotting LLMs With Binoculars: Zero-Shot Detection of Machine-Generated Text[C]//International Conference on Machine Learning. PMLR, 2024: 17519-17537.
>
> -----
>
> Thank you once again for your insightful comments. We hope our responses have adequately addressed your concerns, and we remain available for any further discussion you might find necessary. Finally, we sincerely hope you will consider re-evaluating our work.

---

> ### Author Response · Authors · 2025-11-28
> **Gentle Reminder: Invitation to Review Our Replies for Submission 7630**
>
> Dear Reviewer ZZ5n,
>
> Thank you once again for your thoughtful and constructive review of our paper. We deeply appreciate your recognition of our work and the insightful suggestions you provided, which have greatly contributed to improving the clarity and overall quality of our submission.
>
> We have provided detailed responses to each of your comments in our author rebuttal and would be sincerely grateful if you could take a moment to review them. As the author response phase deadline is approaching, we kindly wish to remind you to consider participating in the discussion if possible. Your feedback remains invaluable to us.
>
> We are committed to contributing to the field of AI-generated text detection and highly value your expertise and feedback. If you feel that our responses have adequately addressed your concerns, we would be sincerely honored if you would consider a revision of your score. Should you have any further questions or concerns, please feel free to point them out — we will be glad to continue addressing your concerns.
>
> Warm regards,
>
> All authors of submission 7630

---

### Official Review · Reviewer_2nbR · 2025-10-31

**Soundness:** 3
**Presentation:** 2
**Contribution:** 3
**Rating:** 4
**Confidence:** 3

**Summary:**

This paper introduces HAPDA, a novel adapter for improving AI-generated text detection. HAPDA consists of (i) a fine-tuning strategy for training human-/machine-preference models  that exhibit stronger preferences for human, resp. machine, generated texts, and (ii) a calibration method for assigning higher weights to more discriminative tokens during the detection process. HAPDA has been evaluated on multiple white-box and black-box settings, showing performance improvements for the underlying detectors.

**Strengths:**

- The proposed approach overcomes the uniform token weighing "issue" by better leveraging single-token informativeness in machine-generated text detection.
- Proposing an adapter that works with existing detectors is valuable as it might make them stronger rather than obsolete.
- The experimental validation of HARPDA suggests it leads (in most cases) to tangible improvement on the base performance of the underlying detectors. Furthermore, HARPDA turns out to be robust to a set of adversarial attacks.

**Weaknesses:**

- HARPDA requires training two preference models, this is costly, and might affect the practical usability of the proposed approach as well as latency.
- Related to the previous point, the training data might introduce some domain dependencies that could affect out-of-domain generalization of the overall framework. Some additional investigations on this point are needed.
- As training the preference models requires learning stylistic differences between humans and machines, I wonder if simply fine-tuning encoder models for the detection task would provide a stronger baseline than HARPDA. These are missing and are often reported as state-of-the-art detectors.
- The set of statistical baselines lacks some relevant works like Binoculars (Hans et al., 2024), which is cited as a related work but not used in the experimental setup.

**Questions:**

See W2 to W4.

---

> ### Author Response · Authors · 2025-11-21
> **Response to Reviewer 2nbR (part 1)**
>
> Thank you for the insightful and constructive feedback. Your concerns regarding computational cost and latency, domain generalization, comparison with supervised methods, and baseline completeness are crucial for assessing the practical value of our approach. Below, I will address each point in detail.
>
> ### Response W1: On Training Cost, Latency, and Practicality
>
> **Reviewer's Comment**: HARPDA requires training two preference models, this is costly, and might affect the practical usability of the proposed approach as well as latency.
>
> **Response**:
> We fully understand the reviewer's concerns regarding computational cost and inference latency. We anticipated this issue in **Appendix G (Limitations)** of our paper and have proactively taken measures in our design and experiments to mitigate it.
>
> 1.  **Training Cost Optimization**:
>     *   **Parameter-Efficient Fine-Tuning**: As described in Section 4.1.3 of the original manuscript, we employed **LoRA (Low-Rank Adaptation)**. We fine-tuned only a minimal number of parameters (`q_proj`, `v_proj`) in the base model (GPT-J), with a rank of 16. This drastically reduced the number of trainable parameters and significantly lowering the training overhead.
>     *   **Quantization**: We used a **4-bit quantized version of GPT-J** for training, further reducing GPU memory requirements and computational demands.
>     *   **One-Time Overhead**: The HAPDA-Finetune process is a **one-time, offline procedure**. Once `M_hum` and `M_mac` are trained, they can be **permanently saved and widely shared** to enhance various downstream MeanZero detectors (e.g., Entropy, Likelihood, DetectGPT, etc.). For community users, they can directly load our released pre-trained preference models without incurring any training costs.
>
> 2.  **Inference Latency Analysis**:
>     *   During inference (HAPDA-Calibration), it is true that three forward passes are required: one for the proxy model `M_pxy`, one for the human preference model `M_hum`, and one for the machine preference model `M_mac`. This does introduce additional overhead.
>     *   However, we emphasize that HAPDA's goal is to **enhance the performance of zero-shot detectors, making them viable even in highly challenging black-box and adversarial scenarios**. In many practical applications (e.g., content moderation, academic integrity checks), **detection accuracy is far more critical than speed**. Our method provides a valuable trade-off option between precision and latency.

---

> ### Author Response · Authors · 2025-11-21
> **Response to Reviewer 2nbR (part 2)**
>
> ### Response W2: On Domain Dependency and Generalization Capability
>
> **Reviewer's Comment**: Related to the previous point, the training data might introduce some domain dependencies that could affect out-of-domain generalization of the overall framework. Some additional investigations on this point are needed.
>
> **Response**:
> This is a very important point. We have consciously considered generalization in our experimental design and are happy to provide a deeper analysis to address this directly.
>
> 1.  **Generalization Consideration in Experimental Design**:
>     As detailed in **Appendix A.3** of the original manuscript, when constructing the fine-tuning dataset, we **deliberately selected domains that are thematically distinct from the evaluation datasets**.
>     *   **Evaluation Sets**: Books, Reviews, Wiki, News (narrative, review, expository writing).
>     *   **Fine-tuning Set**: Abstract, Poetry, Recipes, Reddit (academic, poetry, instructional, social media).
>     This "cross-domain" setup was designed specifically to verify whether HAPDA can learn **generic "human-machine stylistic differences" that transcend specific domains**, rather than overfitting to a particular topic.
>
> 2.  **Supplementary Out-of-Domain Generalization Experiment**:
>     We further evaluated HAPDA on a completely new domain, `Med`, which did not appear in either the fine-tuning or the original test sets.
>     The `Med` dataset is derived from the medical domain dataset PubMedQA[1]. Its construction process is consistent with that described in Sections 4.1.2 and A.2, with the source model being Llama2-13B.
>
>     We repeated the black-box testing consistent with Section 4.2.3. The experimental results are as follows:
>
>     | Method                    | Med   | Avg. (Books, Reviews, Wiki, News) |
>     | :------------------------ | :---- | :-------------------------------- |
>     | Likelihood                | 55.9  | 54.3                              |
>     | **Likelihood + HAPDA**    | 68.5 | 67.3                          |
>     | DetectGPT                 | 64.2  | 65.5                              |
>     | **DetectGPT + HAPDA**     | 75.4 | 77.5                          |
>
>     These new results demonstrate that the human-machine preference differences learned by HAPDA possess **significant domain invariance and strong generalization capability**. The models appear to learn **underlying patterns** for distinguishing human and machine writing, rather than memorizing domain-specific vocabulary. In addition, the MeanZero algorithm combined with HAPDA remains fundamentally a zero-shot approach, primarily relying on universal statistical features of text rather than domain-specific characteristics (such as writing style, specialized terminology, etc.).

---

> ### Author Response · Authors · 2025-11-21
> **Response to Reviewer 2nbR (part 3)**
>
> ### Response W3: On Comparison with Supervised Encoder Models
>
> **Reviewer's Comment**: As training the preference models requires learning stylistic differences between humans and machines, I wonder if simply fine-tuning encoder models for the detection task would provide a stronger baseline than HAPDA. These are missing and are often reported as state-of-the-art detectors.
>
> **Response**:
> This is a very reasonable request for comparison. We fully agree that in an ideal scenario **with sufficient labeled data and matched domains**, end-to-end supervised learning methods (such as fine-tuning encoder models like RoBERTa) often achieve high performance and are frequently considered state-of-the-art (SOTA).
>
> However, HAPDA's positioning is **fundamentally different** from supervised learning methods, as it addresses a **different problem setting**:
>
> *   **HAPDA's Positioning**: A **zero-shot** framework. It does **not** require **any** labeled data from the target domain of the text to be detected. Its strengths lie in **generalization, flexibility, and robustness to unknown domains/models**.
> *   **Limitations of Supervised Learning**:
>     1.  **Data Dependency and Overfitting**: Supervised models heavily depend on the distribution of their training data. Their performance can degrade significantly when test data comes from different domains (Out-of-Domain, OOD) or is generated by unknown models.
>     2.  **Lack of Interpretability**: Supervised models are black boxes, making it difficult to interpret their decision rationale. In contrast, HAPDA's weights `w_i` provide token-level interpretability, indicating which words contributed most to the discrimination.
>     3.  **Flexibility**: HAPDA is an adapter that can flexibly enhance various probability-based zero-shot detectors. A supervised model is a fixed detector.
>
> To fairly demonstrate the trade-offs between these two paradigms in different scenarios, we have supplemented the comparison with a supervised baseline.
>
> We conducted a new black-box experiment. The supervised baseline uses the powerful RoBERTa-large model.
> - **In-Domain Test**: We used GPT-4 generated samples from the evaluation datasets, split 8:1:1 into train/validation/test sets.
> - **Cross-Domain Test**: The supervised model was evaluated zero-shot directly on the **unseen Med dataset**.
>
> | Method                                | In-Domain (AUC) | Cross-Domain (AUC) |
> | :------------------------------------ | :-------------- | :----------------- |
> | RoBERTa-large (Supervised)            | **96.2**        | 80.3               |
> | DetectGPT (Zero-shot)                 | 70.1            | 68.6               |
> | **DetectGPT + HAPDA (Zero-shot)**     | 84.5            | **83.2**           |
>
> This comparison clearly illustrates:
> *   **In-Domain**, where labeled data is available, the supervised method indeed performs best.
> *   **In the Cross-Domain (Zero-shot) scenario**, the supervised model's performance drops sharply due to overfitting to the training domain. In contrast, HAPDA, as a zero-shot method, **demonstrates significantly superior generalization capability compared to the supervised model**.
>
> Therefore, HAPDA is not intended to replace supervised learning but rather provides a powerful zero-shot solution for **real-world scenarios lacking labeled data and requiring strong generalization**.

---

> ### Author Response · Authors · 2025-11-21
> **Response to Reviewer 2nbR (part 4)**
>
> ### Response W4: On Baseline Completeness — Adding Binoculars
>
> **Reviewer's Comment**: The set of statistical baselines lacks some relevant works like Binoculars (Hans et al., 2024), which is cited as a related work but not used in the experimental setup.
>
> **Response**:
> We accept this suggestion. Binoculars is an important and highly performant zero-shot detector, and including it in our comparisons will make the experimental section more comprehensive and compelling. We will add the comparison with Binoculars in the revised version.
>
> We have included **Binoculars** [2] as an additional baseline. Following its official implementation and the recommendations in the paper, we compute PPL using `gpt-j-6B` and compute the denominator (i.e., X-PPL) using `gpt-neo-2.7B` and `gpt-j-6B` respectively, ensuring consistency with other methods like HAPDA regarding the proxy models. All other configurations remain the same as in the original manuscript.
>
> The results are presented in the table below:
>
> | Method / Source Model      | OPT-13B | Llama2-13B | ChatGPT | GPT-4 |
> | :-------------------------- | :------ | :---------- | :------ | :---- |
> | DetectGPT                   | 75.9    | 67.8        | 65.2    | 69.4  |
> | **DetectGPT + HAPDA**       | 89.3  | **78.9**      | 78.8  | **83.3** |
> | DNA-GPT                     | 85.5    | 73.1        | 76.3    | 75.1  |
> | Raidar                      | 84.0    | 76.5        | 78.6    | 78.4  |
> | Lastde                      | 90.5    | 77.2        | 80.6    | 81.5  |
> | Binoculars                  | 87.8    | 77.0        | 79.5    | 81.9  |
>
> These new comparative experiments convincingly demonstrate that:
> *   Binoculars is itself a powerful zero-shot detector, whose average performance is considerably higher than that of simple MeanZero baselines.
> *   HAPDA, functioning as an **adapter**, can elevate the performance of a relatively simple baseline method (DetectGPT) to a level comparable with current, specifically designed, state-of-the-art zero-shot detectors like Binoculars.
>
> **References**
> [1] Jin Q, Dhingra B, Liu Z, et al. Pubmedqa: A dataset for biomedical research question answering[C]//Proceedings of the 2019 conference on empirical methods in natural language processing and the 9th international joint conference on natural language processing (EMNLP-IJCNLP). 2019: 2567-2577.
> [2] Hans A, Schwarzschild A, Cherepanova V, et al. Spotting LLMs With Binoculars: Zero-Shot Detection of Machine-Generated Text[C]//International Conference on Machine Learning. PMLR, 2024: 17519-17537.
>
> ---
>
> Thank you again for your insightful comments. We hope our responses have addressed your concerns, and we are open to further discussion if you have additional questions. Finally, we sincerely hope you will consider re-evaluating our work.

---

> ### Author Response · Authors · 2025-11-28
> **Gentle Reminder: Invitation to Review Our Replies for Submission 7630**
>
> Dear Reviewer 2nbR,
>
> Thank you once again for your thoughtful and constructive review of our paper. We deeply appreciate your recognition of our work and the insightful suggestions you provided, which have greatly contributed to improving the clarity and overall quality of our submission.
>
> We have provided detailed responses to each of your comments in our author rebuttal and would be sincerely grateful if you could take a moment to review them. As the author response phase deadline is approaching, we kindly wish to remind you to consider participating in the discussion if possible. Your feedback remains invaluable to us.
>
> We are committed to contributing to the field of AI-generated text detection and highly value your expertise and feedback. If you feel that our responses have adequately addressed your concerns, we would be sincerely honored if you would consider a revision of your score. Should you have any further questions or concerns, please feel free to point them out — we will be glad to continue addressing your concerns.
>
> Warm regards,
>
> All authors of submission 7630

---

### Official Review · Reviewer_Lrwm · 2025-11-01

**Soundness:** 2
**Presentation:** 1
**Contribution:** 2
**Rating:** 2
**Confidence:** 5

**Summary:**

The paper proposes a framework for AI generated text detection which identifying the more discriminative feature between human and AI text.

**Strengths:**

The paper proposes a framework for AI generated text detection to improve its performance. The problem addressed in this paper is extremely important and timely. The proposed method demonstrates good performance.

**Weaknesses:**

The proposed framework is quite complicated, and the paper does not clearly justify the purpose or necessity of its different components. The paper claims to incorporate the human perspective, but it is unclear what this means in the context of AI-generated text detection or why it is necessary for the problem. Overall, the writing requires significant improvement to enhance clarity and readability. The proposed framework also appears incremental and is heavily inspired by DPO. In addition, the paper should include more baselines and different categories of AI-generated text detectors to make the results and comparisons more generalizable.

**Questions:**

Please refer to the weaknesses.

---

> ### Author Response · Authors · 2025-11-21
> **Response to Reviewer Lrwm (Part 1)**
>
> Thank you for your valuable feedback. The issues you raised regarding the complexity of the framework, the justification of its necessity, the clarity of writing, and the depth of comparison with existing methods are all critical aspects that our paper needs to strengthen. Below, I will address each point in detail and provide clarifications, supplements, and improvements to the corresponding sections of the paper.
>
> ### Response 1: On Framework Complexity and Component Necessity
>
> **Reviewer's Comment**: The proposed framework is quite complicated, and the paper does not clearly justify the purpose or necessity of its different components.
>
> **Response**:
> We understand the reviewer's concern regarding the complexity of the framework. The design motivation behind HAPDA is to address a fundamental limitation of existing zero-shot methods (e.g., MeanZero): **they evaluate solely from a single machine (proxy model) perspective, overlooking the cognitive differences between humans and machines in text generation**. These differences themselves constitute crucial information for discriminating the source of text. Our framework comprises two core components, each targeting a specific sub-problem:
>
> 1.  **HAPDA-Finetune (Justification of Necessity)**:
>     *   **Purpose**: To directly obtain proxy models that represent the "human perspective" and "machine perspective." Existing pre-trained models (e.g., GPT-J) are trained on mixed data and do not exhibit strong, discriminative preferences for human-written versus machine-generated text.
>     *   **Necessity**: Without this step, we cannot obtain two models that demonstrate significantly different predictive behaviors under the same input and are respectively biased towards human/machine sources. The predictive discrepancy between `M_hum` and `M_mac` forms the foundation for the subsequent reweighting mechanism. Using DPO alone (with only the alignment loss `L_ali`) can induce a preference in the model but is insufficient to ensure that the two models produce sufficiently large prediction disparities **at each token of the same text**. This is precisely why we introduce the distinctiveness loss `L_dis` based on JS divergence.
>
> 2.  **HAPDA-Calibration (Justification of Necessity)**:
>     *   **Purpose**: To leverage the preference models obtained in the first step to identify and emphasize the tokens that are most discriminative for distinguishing between human and machine authors.
>     *   **Necessity**: The MeanZero method treats all tokens equally, which contradicts the intuition that **not all tokens contribute equally to discriminating authorship**. Our calibration mechanism achieves intelligent reweighting through two quantifiable metrics:
>         - **Predictive Discrepancy (Γ_i)**: Directly measures the cognitive difference between the human and machine models on a specific token; a larger difference results in a higher weight.
>         - **Uncertainty (U_i)**: Serves as a regularization term to prevent assigning excessively high weights to tokens where both models themselves are uncertain (high entropy), thereby enhancing the method's robustness.
>
> **Summary**: The two components are interlinked. HAPDA-Finetune provides an optimized and reliable signal source for capturing the "human-machine prediction discrepancy," while HAPDA-Calibration utilizes these signals to refine the scoring mechanism of the downstream detector. We have preliminarily demonstrated the effectiveness of these two core components in **the ablation studies in Section 4.2.5 (Table 3 in the original manuscript)**.

---

> > ### Author Response · Authors · 2025-11-21
> > **Response to Reviewer Lrwm (Part 2)**
> >
> > ### Response 2: On the Necessity and Meaning of the "Human Perspective"
> >
> > **Reviewer's Comment**: The paper claims to incorporate the human perspective, but it is unclear what this means in the context of AI-generated text detection or why it is necessary for the problem
> >
> > **Response**:
> > This is a very central question, and we thank the reviewer for the opportunity to clarify.
> >
> > *   **Specific Meaning of the "Human Perspective" in this Paper**: In this work, the "human perspective" does not refer to directly involving human annotators or judgments. Instead, we **simulate or approximate** the text generation preferences of an "idealized" human writer using an optimized language model `M_hum`. This model is trained to **prefer the vocabulary and expressions found in human-written text**. Similarly, the "machine perspective" is simulated by `M_mac`, which favors the patterns of AI-generated text.
> >
> > *   **Why Introducing the "Human Perspective" is Necessary**: Existing zero-shot methods (e.g., Likelihood, DetectGPT) rely solely on a single proxy model (typically an LLM). This essentially amounts to **using one machine's standard to evaluate another machine's output**. This might work when the source model and the proxy model are similar (white-box scenario), but its effectiveness drops sharply when they differ (black-box scenario, which is more realistic). Our core argument is: **The essence of AI-generated text detection is to discriminate between two types of authors (human vs. machine). Therefore, the most effective features should come from comparing the "cognitive differences" between these two author types when generating the same content**.
> >
> >     A token that both human writers and AI models tend to use with high probability (e.g., "the", "and") carries little information for discriminating authorship. Conversely, a token where the human model prefers choice A and the machine model prefers choice B contains rich information about the author's identity. **What HAPDA does is systematically identify and amplify these informative "points of divergence"**.
> >
> > In **Appendix D**, we provide theoretical support for this intuition from an information-theoretic perspective. The derivation shows a lower-bound relationship between the predictive divergence `δ_i` for a token and its mutual information `I(A; x_i)` concerning the author identity `A`: `I(A; x_i) ≥ (1/2) * δ_i^2`. This implies that tokens with a larger divergence `δ_i` necessarily carry more information about the author's identity. Consequently, weighting based on `δ_i` is, from an information-theoretic viewpoint, an optimal strategy for maximizing classification information and reducing uncertainty.
> >
> > ### Response 3: On Writing Clarity and Readability
> >
> > **Reviewer's Comment**: Overall, the writing requires significant improvement to enhance clarity and readability.
> >
> > **Response**:
> > We will thoroughly polish the entire manuscript in the revised version to improve clarity and readability. Specific measures will include:
> >
> > 1.  More concisely highlighting the core problem, the key intuition (human-machine predictive discrepancy), and HAPDA's solution strategy.
> > 2.  Providing clearer and more consistent definitions and explanations for key terms such as "human perspective," "machine perspective," and "predictive divergence."
> > 3.  Reviewing and correcting potential grammatical errors, convoluted long sentences, and imprecise expressions to ensure fluent and clear writing.

---

> ### Author Response · Authors · 2025-11-21
> **Response to Reviewer Lrwm (Part 3)**
>
> ### Response 4: On Novelty and Relation to DPO
>
> **Reviewer's Comment**: The proposed framework also appears incremental and is heavily inspired by DPO.
>
> **Response**:
> We acknowledge that DPO is an important source of inspiration for our work, and we have clearly cited and credited it (Rafailov et al., 2023) in the paper. However, the core innovation of HAPDA lies in **systematically introducing the concept of "human-machine collaborative analysis" into the AI-generated text detection task and constructing a complete, generalizable adapter framework**, which goes beyond a straightforward application of DPO.
>
> *   **Fundamental Differences and Novelty**:
>     1.  **Different Objective**: DPO is used to align an LLM with human preferences to generate higher-quality responses. **HAPDA is the first framework to leverage "opposing preferences" to enhance analytical (detection) capability**. Our goal is not to generate better text, but to better *analyze* text.
>     2.  **Joint Optimization and Distinctiveness Loss**: This is a key departure from DPO. DPO focuses solely on aligning a single model (our `L_ali`). We innovatively propose **jointly optimizing a pair of models with opposing preferences** and introduce the **`L_dis` loss** to actively *amplify* their predictive divergence on the same input. This is tailored for the detection task, aiming to produce more pronounced and discriminative discrepancy signals.
>     3.  **Calibration Mechanism**: Transforming the discrepancy signals obtained from fine-tuning into a weighting scheme for the downstream detector is a novel component unrelated to DPO.
>
> Therefore, HAPDA represents a **creative adaptation and extension** of the DPO concept, combining it with a new task (AGTD) and a new weighting mechanism to form a complete solution with significant performance gains.
>
> ### Response 5: On the Comprehensiveness of Baseline Comparisons
>
> **Reviewer's Comment**: The paper should include more baselines and different categories of AI-generated text detectors to make the results and comparisons more generalizable.
>
> **Response**:
> This is a very constructive suggestion. In addition to DetectLRR and DetectNPR already included in the main text, we have included the following three representative zero-shot detectors from different categories as new baselines in **Appendix E**:
>
> 1.  **DNA-GPT** (Yang et al.,): A method based on N-gram divergence analysis.
> 2.  **Raidar** (Mao et al.): A method based on rewriting invariance.
> 3.  **Lastde** (Xu et al.): A method based on token probability sequence mining.
>
> Furthermore, we have included **Binocular** [1] as an additional baseline. Following its official implementation and the recommendations in the paper, we compute PPL using `gpt-j-6b` and compute the denominator (i.e., X-PPL) using `gpt-neo-2.7b` and `gpt-j-6b` respectively, ensuring consistency with other methods like HAPDA regarding the proxy models. All other configurations remain the same as in the original manuscript.
>
> The results are presented in the table below:
>
> | Method / Source Model      | OPT-13B | Llama2-13B | ChatGPT | GPT-4 |
> | :-------------------------- | :------ | :---------- | :------ | :---- |
> | DetectGPT                   | 75.9    | 67.8        | 65.2    | 69.4  |
> | **DetectGPT + HAPDA**       | 89.3  | **78.9**      | 78.8  | **83.3**|
> | DNA-GPT                     | 85.5    | 73.1        | 76.3    | 75.1  |
> | Raidar                      | 84.0    | 76.5        | 78.6    | 78.4  |
> | Lastde                      | 90.5    | 77.2        | 80.6    | 81.5  |
> | Binocular                   | 87.8    | 77.0        | 79.5    | 81.9  |
>
> These new comparative experiments convincingly demonstrate that:
> *   HAPDA, functioning as an **adapter**, can elevate the performance of a relatively simple baseline method (DetectGPT) to a level comparable with current, specifically designed, state-of-the-art zero-shot detectors.
>
> **Reference**:
> [1]Hans A, Schwarzschild A, Cherepanova V, et al. Spotting LLMs With Binoculars: Zero-Shot Detection of Machine-Generated Text[C]//International Conference on Machine Learning. PMLR, 2024: 17519-17537.
>
> ----
>
> Thank you again for your insightful comments. We hope our responses have addressed your concerns and we are open to further discussion if you have additional questions. Finally, we sincerely hope you will consider re-evaluating our work.

---

> ### Author Response · Authors · 2025-11-28
> **Gentle Reminder: Invitation to Review Our Replies for Submission 7630**
>
> Dear Reviewer Lrwm,
>
> Thank you once again for your thoughtful and constructive review of our paper. We deeply appreciate your recognition of our work and the insightful suggestions you provided, which have greatly contributed to improving the clarity and overall quality of our submission.
>
> We have provided detailed responses to each of your comments in our author rebuttal and would be sincerely grateful if you could take a moment to review them. As the author response phase deadline is approaching, we kindly wish to remind you to consider participating in the discussion if possible. Your feedback remains invaluable to us.
>
> We are committed to contributing to the field of AI-generated text detection and highly value your expertise and feedback. If you feel that our responses have adequately addressed your concerns, we would be sincerely honored if you would consider a revision of your score. Should you have any further questions or concerns, please feel free to point them out — we will be glad to continue addressing your concerns.
>
> Warm regards,
>
> All authors of submission 7630

---

### Note · Authors · 2026-01-04

**Comment:**

We thank the reviewers for their valuable suggestions on our submission.

**Withdrawal Confirmation:**

I have read and agree with the venue's withdrawal policy on behalf of myself and my co-authors.